# Comment on "Review of Experimental Studies of Secondary Ice Production" by Korolev and Leisner (2020)

By Vaughan T. J. Phillips[1], Jun-Ichi Yano[2], Akash Deshmukh[1] and Deepak Waman[1]

[1] Department of Physical Geography, University of Lund, Lund, Sweden

[2] CNRM, UMR 3589 (CNRS), Météo France, 31057 Toulouse Cedex, France

*Correspondence to*: Vaughan Phillips (Vaughan.phillips@nateko.lu.se)

**Abstract.** This is a comment on the review by Korolev and Leisner (2020, hereafter 'KL2020'). The only two laboratory/field studies ever to measure breakup in ice-ice collisions for in-cloud conditions were negatively criticised by KL2020, as were our subsequent theoretical and modelling studies informed by both studies. Firstly, hypothetically even without any further laboratory experiments, such theoretical and modelling studies would continue to be possible, being based on classical mechanics and statistical physics. They are not sensitive to the accuracy of lab data for typical situations, partly because nonlinear explosive growth of ice concentrations continues until some maximum concentration is reached. To a degree, the same final concentration is expected regardless of the fragment number per collision. Secondly, there is no evidence that both lab/field observational studies characterising fragmentation in ice-ice collisions are either mutually conflicting or erroneous such that they cannot be used to represent this breakup in numerical models, contrary to the review. The fact that the ice spheres of one experiment were hail-sized (2 cm) is not a problem if a universal theoretical formulation such as ours with fundamental dependencies is informed by it. Although both lab/field studies involved head-on collisions, rotational kinetic energy for all collisions generally is only a small fraction of the initial collision kinetic energy (CKE) anyway. Although both lab/field experiments involved fixed targets, that is not a problem since fixing of the target is represented *via* CKE in in any energy-based formulation such as ours. Finally, scaling analysis suggests that breakup of ice during sublimation can make a significant contribution to ice enhancement in clouds, again contrary to the impression given by the review.

## 1 Introduction

The literature of secondary ice production (SIP) was recently reviewed by KL2020. The focus of their review is on laboratory experiments. It is commendable that their review attempts to re-invigorate laboratory observations of the various types of fragmentation of ice, research that has in the last decade been oriented chiefly towards heterogeneous ice nucleation. We agree with KL2020 that future experiments about breakup of ice can only enhance the accuracy of its empirical characterisation.

Regarding observations of breakup in ice-ice collisions for plausible in-cloud conditions, only two prior lab/field studies have ever been published quantifying it, as far as we are aware (Vardiman 1978; Takahashi *et al.* 1995).

However, the unfortunate impression is given to the reader that numerical modeling and theoretical studies of breakup in ice-ice collisions are currently impossible due to the fact that reliable data for laboratory experiments are somehow critically missing. Even if the available data were unreliable or unrepresentative of natural clouds, which is actually not the case (Sec. 4), our stand-alone theoretical and modelling studies would nevertheless still have been possible, being based on classical mechanics of collisions and statistical physics:

- Yano and Phillips (2011) and Yano *et al.* (2016) delineated the time-scale of explosive growth of ice concentrations and the conditions for such instability in terms of dimensionless parameters that provide a phase-space of the system.
- The role of stochastic variability among fragmentation events in promoting the onset of instability is elucidated theoretically by means of the Fokker–Planck equation (Yano and Phillips 2016).
- Phillips *et al.* (2017a) show a formulation based on principles of energy conservation and statistical variability of strengths of asperities, using a recent definition of the coefficient of restitution. An application of this versatile framework to a real cloud situation involved constraining some parameters with the lab/field data (Vardiman 1978; Takahashi *et al.* 1995) but exists as a theory independently of such data.

When KL2020 were negatively criticizing both lab/field studies (Vardiman 1978; Takahashi *et al.* 1995), they mentioned three of these studies in the same context.

The purpose of our commentary is to point out some misunderstandings apparent in the review by KL2020. The focus is on Section 4 of KL2020, which covered the topic of breakup in ice-ice collisions, where we have contributed both in theories and modelling. KL2020 have issued a Corrigendum for how their section dealt with our three earlier studies about fragmentation of ice (Yano and Phillips 2011; Yano *et al.* 2016; Phillips *et al.* 2017a).

The structure of the commentary is as follows. First, to enable the reader to follow the technical details of our arguments defending the lab/field studies criticised by KL2020, the next two sections summarise their observations of breakup in ice-ice collisions and how these inspired our subsequent theoretical and modeling contributions. That is followed by specific comments on the review by KL2020 (their Section 4) and its Corrigendum in Sec. 4. This describes how the review by KL2020 made several misunderstandings in their survey of those lab studies underpinning our modeling and theoretical studies of breakup. We describe these errors that their Corrigendum has not rectified.

(a)

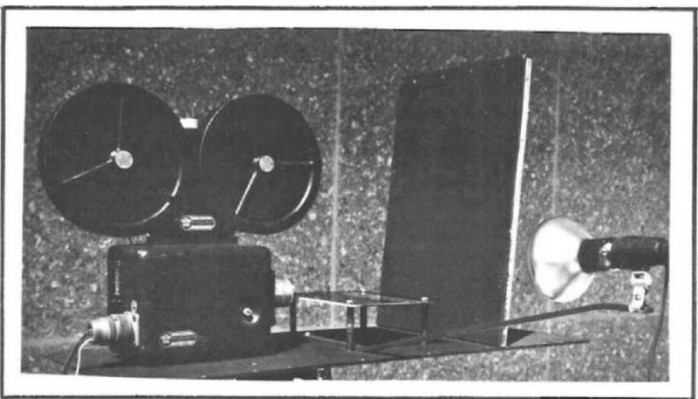

(b)

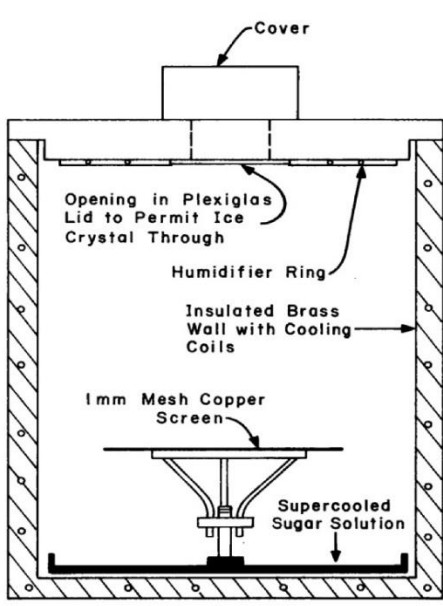

**Figure 1:** *The portable instruments used to determine the number of fragments from natural falling ice particles. They were placed outdoors on the ground with natural ice falling into them. Field observations in winters of 1973-74 were made with the simplified setup in (a), with a l6mm movie camera, a black plexiglass plate and an optically-black background. This instrument had been previously verified as realistic by use of the portable chamber in (b) with ice particles falling through the hole in the top and into the chamber, breaking on impact with the copper screen. Whereas in (a) fragments were counted only by inspection of the video footage, they were counted in (a) by eye after falling through the mesh into the supercooled solution. From Vardiman (1978). © American Meteorological Society. Used with permission.*



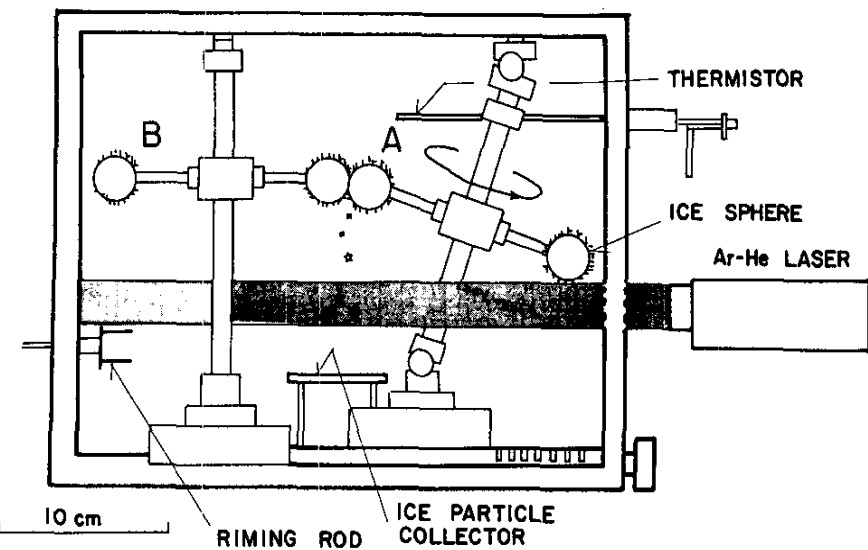

**Figure 2**: *Experimental apparatus of Takahashi et al. (1995) to observe fragmentation in ice-ice collisions: (A) ice spheres rotated at a tangential speed of 4 m s$^{-1}$, while (B) ice spheres were stationary. The ejected ice particles were collected on the plate below. Cloud droplets were supplied from the centre of the right wall. From Takahashi et al. (1995). © American Meteorological Society. Used with permission.*

## 2 Published Laboratory/Field Observations of Breakup in Ice-Ice Collisions

A flaw of the review by KL2020 is the claim that the only two lab/field studies observing fragmentation in ice-ice collisions published hitherto are somehow so erroneous that they cannot provide any basis for representing this type of SIP in numerical models. Before our counter-arguments defending their utility (Sec. 4.1), salient features of both lab/field studies are summarised.

First, Vardiman (1978) constructed a portable instrument deployed outdoors on mountainsides in USA during events of ice precipitation (Fig. 1a). Ice particles fell through the field of view of a movie camera onto a plexiglass fixed plate and fragments were counted later by inspection of the film. Fragment counts had earlier been verified as accurate by inter-comparison with an alternative portable chamber that captured all fragments (Fig. 1b). The chamber had an opening at the top through which natural ice particles (about 0.5 to 5 mm in diameter) fell, descending into the field of view of a camera and impacting a copper

mesh. All fragments were captured in a supercooled solution and counted by eye. The numbers of fragments per collision were measured (Fig. 1a) for various types of ice particle:

- Unrimed dendrites
- Lightly/moderately rimed dendrites
• Heavily rimed dendrites
- Lightly/moderately rimed spatial crystals
- Graupel

For each morphological type, the number of fragments per collision was mostly in the range of 1 to 100 for most sizes,
increasing with size. This number was published as a function of momentum change on the fixed plate (Vardiman 1978).

Second, Takahashi *et al.* (1995) observed collisions between two giant spheres of ice (2 cm diameter), one rimed (A) and the other unrimed (B), in a cold-box in a laboratory. Figure 2 shows how these were performed. The unrimed sphere was fixed while the rimed one was on a rotating arm. Both particles were intended to be representative of small and large graupel
colliding in a real cloud after falling into regions of weak liquid water content (LWC), where the smaller one grows by vapour deposition predominantly and the other by riming, as had been observed by an airborne video-probe (Takahashi 1993; Takahashi and Kuhara 1993). High concentrations of ice were observed for such collisions by that video-probe aloft. Takahashi *et al*. (1995) in the lab experiment observed hundreds of ice fragments per collision ($N$).

Additionally, Takahashi *et al*. (1995) provided a reduced estimate of $N$ for natural graupel of the more common millimeter-sizes seen in clouds ($N \approx 50$). Takahashi *et al.* (1995) did this rescaling according to the peak collision force for the fallspeed difference at such natural sizes (0.5 and 4 mm). However, our formulation (Phillips *et al*. (2017a) suggests that, when doing that rescaling, Takahashi *et al.* may not have accounted for the reduced area of contact at these more natural sizes, possibly over-estimating of $N$ by an order of magnitude (Sec. 3).

In summary, both lab/field studies encompass a variety of types of ice morphology and energies of impact. Although perhaps incomplete in terms of representing the full extent of this variety in natural clouds (Sections 4, 5), the alternative option of ignoring such breakup when constructing cloud models would seem absurd in view of the prolific fragmentation they observed. Ice-ice collisions are the norm, not the exception, in cold clouds (e.g. snowflakes are aggregates of crystals).

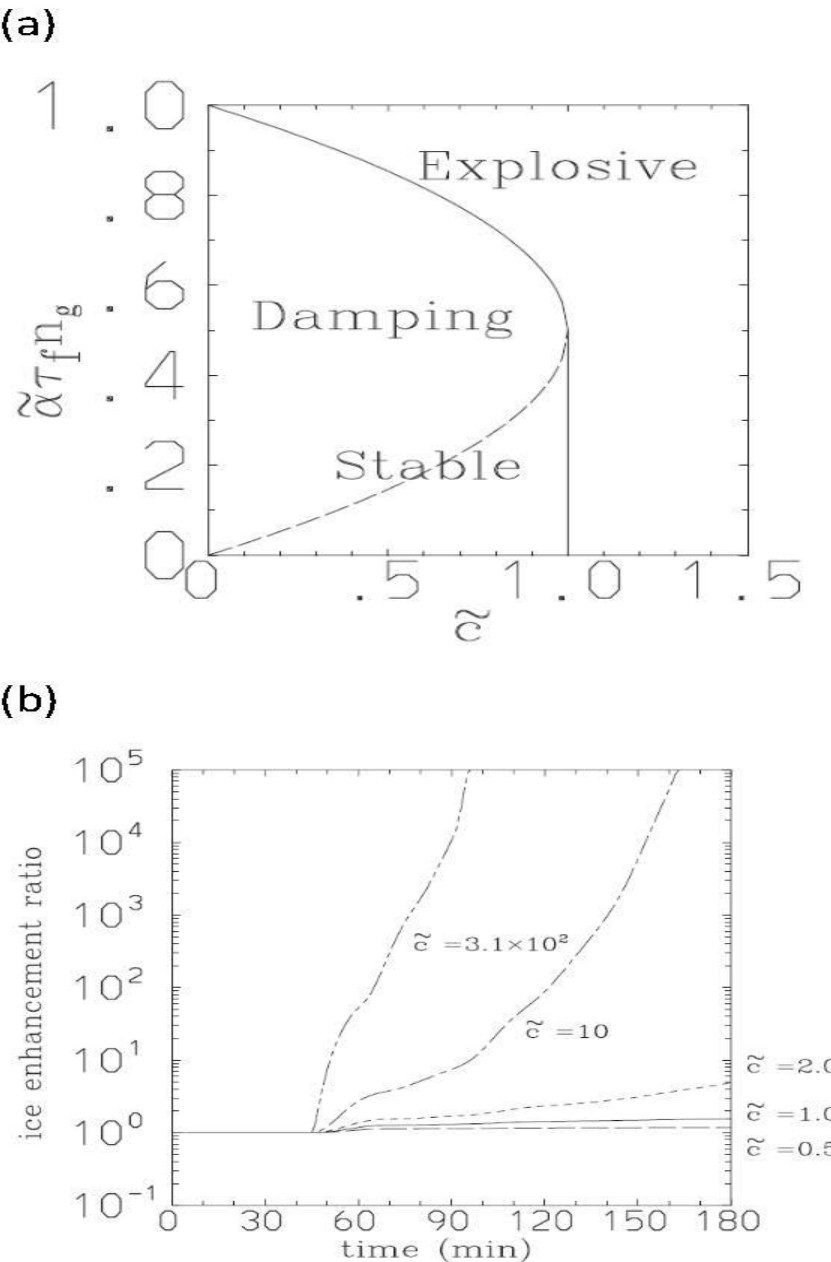

**Figure 3:** (a) The phase-space of stability for the 0D model of ice-ice breakup for a population of ice crystals, and small and large ice precipitation particles. The multiplication efficiency, $\tilde{c}$, (proportional to the number of fragments per ice-ice collision) and dimensionless initial precipitation concentration are the two axes.   In (b) the time evolution of the ice enhancement is shown for various values of $\tilde{c}$. From Yano and Phillips (2011).

### 3. Our Previous Contributions: Theories and Models Informed by Both Lab/Field Experiments of Collisional Breakup

KL2020 suggest that three of our modeling and theoretical studies are somehow critically dependent on the lab/field studies (Vardiman 1978; Takahashi *et al*. 1995) noted above (Sec. 2). Yet in reality, Yano and Phillips (2011) and Yano *et al*. (2016) treated breakup at the cloud-scale, not the particle scale. Both of our studies provided a 0D analytical model of a cloud, delineating a general theory of the nonlinear growth of ice concentrations with three species of ice: crystals, small graupel and large graupel. It was found that a sole non-dimensional parameter, $\tilde{c}$, which measures the efficiency of ice multiplication, characterises the system in its 2D phase-space (Fig. 3a). $\tilde{c}$ is proportional to the number of fragments per ice-ice collision. A systematic investigation of the behavior of the system is performed by varying this nondimensional parameter. A tendency for explosive ice-multiplication is identified in a regime with $\tilde{c} > 1$.

Although the laboratory experiment by Takahashi *et al.* (1995), (Sec. 2), was referred to in both papers (Yano and Phillips 2011; Yano *et al.* 2016), this was solely for the purpose of estimating the number of ice fragments per collision ($N$), which determines the order of magnitude of the nondimensional parameter, $\tilde{c}$. The obtained estimate for a standard case of deep convection is $\tilde{c} \sim 300$, assuming the estimate rescaled by Takahashi *et al.* ($N \sim 50$). This standard value (300) is almost 3 orders of magnitude higher than the threshold of $\tilde{c}$ for onset of instability with explosive growth of ice concentrations, namely unity (Fig. 3a). In this respect, our theoretical studies do not depend on any results of laboratory experiments in any critical manner.

Figure 3b illustrates how a change of the number of fragments per collision by an order of magnitude only changes the time taken to attain an ice enhancement ratio of 100 by about 30 mins (comparing $\tilde{c} \sim 300$ with $\tilde{c} \sim 30$), regarding breakup in graupel-graupel collisions. Close to the threshold ($\tilde{c} < 3$), the time-scale becomes longer than for most cloud systems. Phillips *et al.* (2017b) modified the theory of Yano and Phillips (2011) to treat breakup in graupel-snow collisions, estimating that $\tilde{c} \approx 10$. They found this was consistent with an increase by an order of magnitude of average ice concentrations in about an hour, as predicted by the detailed cloud simulation. As the explosion of concentrations of ice proceeds, eventually the critical maximum ice concentration for conversion of mixed-phase cloud to ice-only occurs, with the timing of arrival at this final state being more closely related to the order of magnitude of $\tilde{c}$ rather than to $\tilde{c}$ itself.

Additionally, Yano and Phillips (2016) further considered a contribution of stochastic fluctuations of the ice-fragmentation number by collision. The paper shows that multiplicative noise effect induced by this stochasticity may lead to an explosive multiplication even under a subcritical state (i.e., $\tilde{c} < 1$). The system was characterised by the time-evolution of a probability distribution function in a phase-space according to the Fokker-Plank equation.

An illustration of how our theoretical studies are not critically sensitive to the results of the lab experiments is as follows. Yano and Phillips (2011) and Yano *et al.* (2016) used the original rescaled estimate from Takahashi *et al.* (1995) of $N$ (50 splinters) for the standard case involving collisions among graupel particles of more typical natural sizes (0.5 and 4 mm). If now, in light of the formulation by Phillips *et al.* (2017b), this rescaled estimate ($N \approx 50$) is corrected with a value from our formulation (Phillips *et al.* 2017a) accounting for the reduced area of contact, then an estimate of only a few splinters per collision is obtained for the same pair of natural sizes. This implies that $\tilde{c} \sim 10$, which is still supercritical (Fig. 3a). Conclusions about the overall effects from such breakup on any cloud in terms of feedbacks, instability and dependencies (e.g. on ascent) remain unchanged. Note that such a correction does not affect the actual observations by Takahashi *et al.* (1995), which remain valid.

Next, Phillips *et al.* (2017a) created a formulation for the number of fragments from any collision of two ice particles, as a function of their sizes, velocities, temperature and morphology. It was based on principles of theoretical statistical physics considering energy conservation at the particle-scale. For a pair of colliding particles that initially are not rotating:

$$K_0 = K_1 + \Delta S + K_{th} \qquad (1)$$

Here $K_0$ is the initial collision kinetic energy (CKE), which is defined simply as the total translational kinetic energy of both particles in the frame of their combined centre of mass. Note that CKE is not a concept restricted to head-on collisions. $K_1$ is the final kinetic energy of the system after impact in that same frame of reference, consisting principally of the CKE of the colliding particles, but also their rotational kinetic energy. $\Delta S$ is the change in surface adhesion energy after impact. $K_{th}$ is the energy lost on impact as noise, heat and inelastic deformation including plastic deformation and breakage. A coefficient of restitution defined in terms of the fractional energy loss, $K_{th}/K_0$, from Wall *et al.* (1990), (see also Supulver *et al.* 1995) was applied. This coefficient is not restricted to head-on collisions and includes the possibility of oblique collisions with rotation afterwards. This, together with observed statistics of surface asperities, led to the formulation:

$$N = \alpha A(T, D \dots) \left( 1 - \exp\left[ -\left( \frac{CK_0}{\alpha A(T, D, \dots -)} \right)^{\gamma} \right] \right) \qquad (2)$$

Here, $N$ is the number of fragments per collision and $\alpha$ is surface area (equivalent spherical) of the smaller particle in the colliding pair, while $A(T, D \dots)$, $\gamma$ and $C$ are empirical constants expressing how fragility depends on ice morphology and ambient conditions. Figure 4 shows how the general mathematical form of Eq (2) is consistent with independent data from Vidaurre and Hallett (2009) for a wide range of sizes at a given speed (130 m/s).

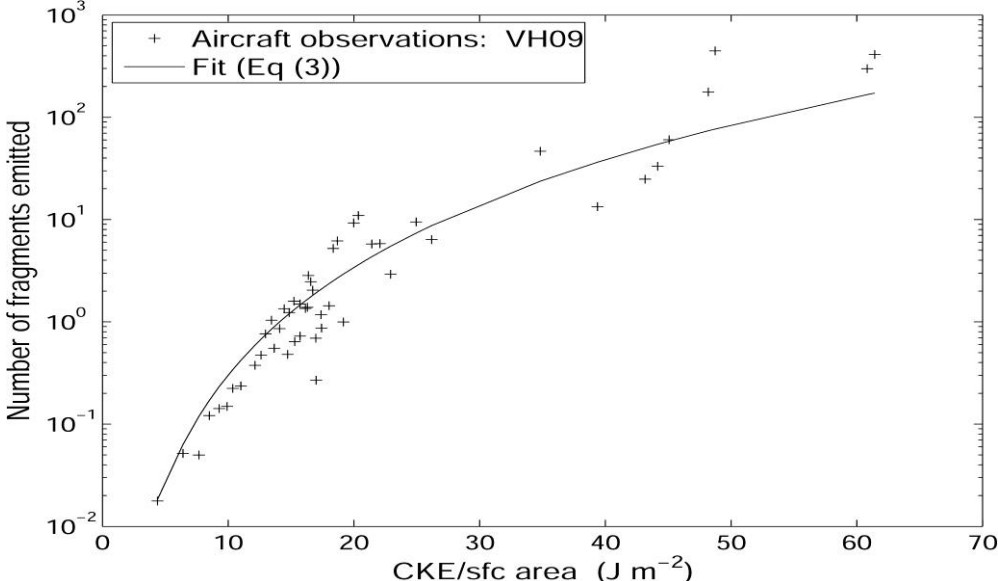

**Figure 4:** *Measurements by Vidaurre and Hallett (2009) of numbers of fragments from ice crystals of many sizes (10–300 µm) impacting the formvar replicator flown at 130 ms$^{-1}$ through clouds above Oklahoma, plotted as a function of the ratio of CKE to surface area of the ice particle (see Eq (2)). From Phillips et al. (2017a). © American Meteorological Society. Used with permission.*

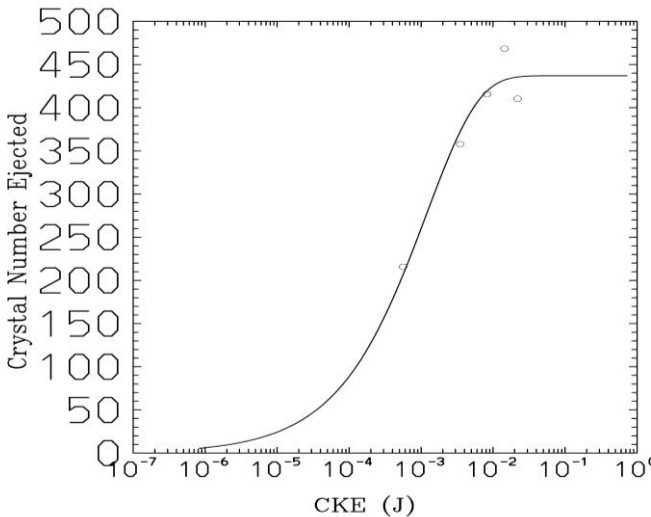

**Figure 5:** *Measurements of numbers of fragments per hail–hail (2-cm diameter) collision (circles), as a function of initial CKE, which we inferred using Hertz theory from the published data of the experiment by Takahashi et al. (1995). From Phillips et al. (2017a).* © American Meteorological Society. Used with permission.

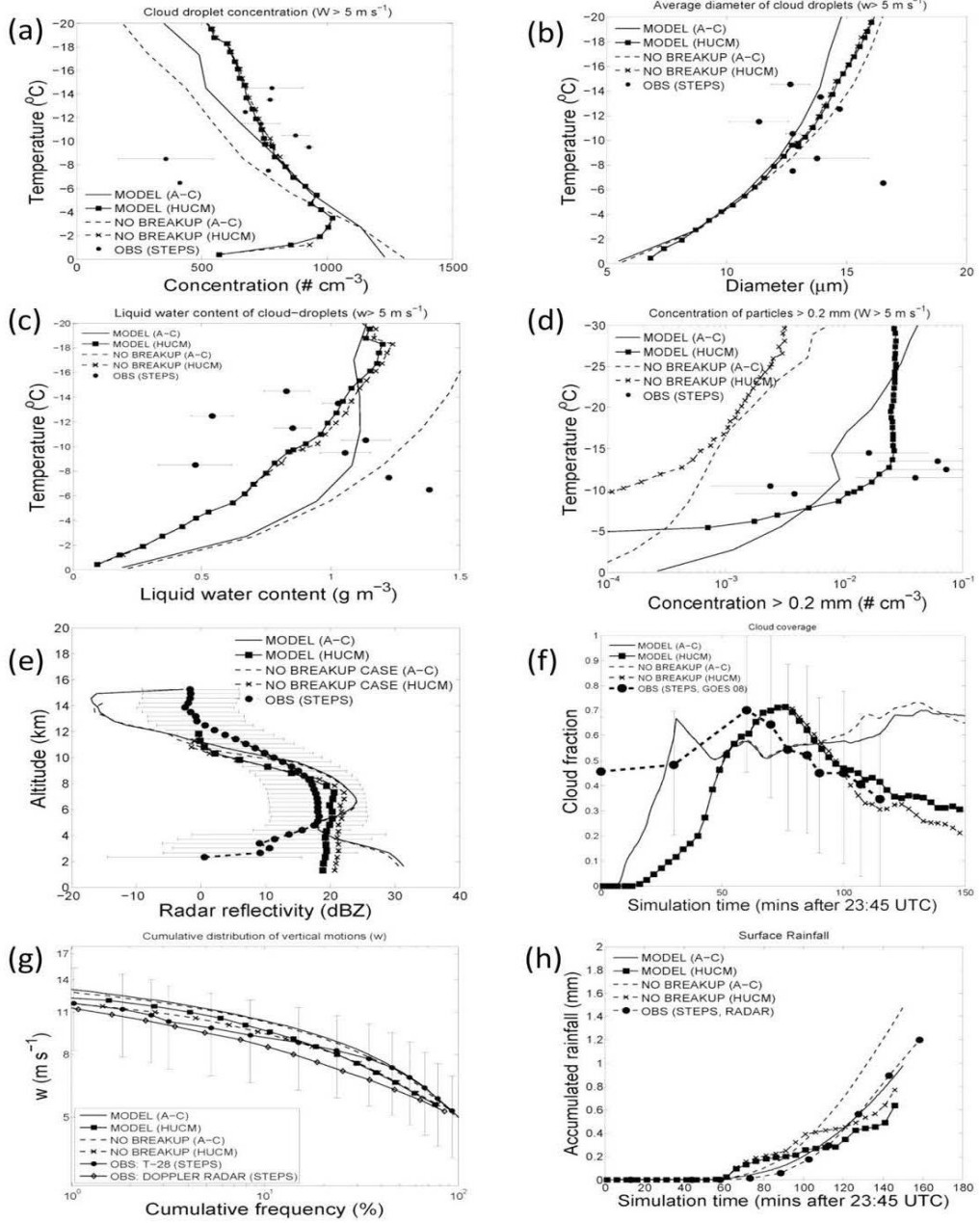

**Figure 6:** Validation of the aerosol-cloud (AC) simulation with and without breakup in ice-ice collisions, using aircraft data, radar data and satellite data. Quantities shown are cloud-droplet concentrations and mean size, liquid water content, ice concentration, radar reflectivity, cloud fraction, ascent statistics and surface accumulated precipitation from (a) to (h). Ice concentration is shown logarithmically in (d), both with and without breakup. Another simulation of the same case by HUCM model is also shown. Reproduced from Phillips *et al*. (2017b). © American Meteorological Society. Used with permission.

The empirical constants of the theory (Eq (2)) were constrained for observations of graupel-graupel collisions over a wide range of CKEs and impact speeds by the lab experiment from Takahashi *et al.* (1995), (Fig. 5). For other microphysical species, these constants were constrained by observations with an instrument (Fig. 1a) outdoors on a mountainside by Vardiman (1978), (Figure 1; Sec. 2). Thus, splintering for each permutation of microphysical species in ice-ice collisions between snow, crystals, graupel and hail was represented by our formulation (Phillips *et al.* 2017a).

Phillips *et al.* (2017b) applied the formulation in the 'aerosol-cloud model' (AC) to quantify the role of ice-ice collisional breakup for a convective storm observed by radar and aircraft over the US High Plains (STEPS). Ice-ice breakup generated over 99% of all non-homogeneously nucleated ice particles, and was needed for agreement of ice concentrations predicted by AC with aircraft observations, which were corrected for artificial shattering biases. Only with our scheme for breakup included

were various observations (e.g. supercooled liquid water content) of the clouds reproduced. This discovery was never mentioned in the review by KL2020.

Curiously, in the budgets of all ice particles initiated throughout the simulated storm, the graupel-snow collisions were orders of magnitude more prolific in generating splinters than collisions among only graupel/hail (Phillips *et al.* 2017b). Thus, Phillips

*et al.* argued that the 0D analytical model of Yano and Phillips (2011) and related theory of instability were directly applicable to graupel-snow collisions, except with the small ice precipitation species defined as "snow" (> 0.3 mm; crystals and their aggregates) instead of "small graupel". Phillips *et al.* (2017b) estimated the corresponding value of multiplication efficiency for graupel-snow collisions as $\tilde{c} \sim 10$ for the cold-based convective storm (STEPS).

There were no tune-able parameters to adjust in that simulation by AC. Two cloud models with contrasting architecture simulated the same case of a mesoscale convective system that was about 100 km wide. Ours was AC with hybrid bin/bulk microphysics, the other was pure spectral microphysics, namely the Hebrew University Cloud Model (HUCM). Both models allowed the same conclusion to be reached: only by including ice-ice collisional breakup could the order of magnitude of the ice concentration observed be reproduced. Thus, there was no possibility of tuning the cloud condensation nucleus (CCN) or

ice nucleus (IN) activity to give the desired ice concentration. Indeed, the vast majority (about 99%) of ice particles initiated in the mixed-phase clouds were secondary and there was little activity of any (CCN-sensitive) warm rain process that might yield graupel by raindrop-freezing. Equally, the ascent statistics observed by aircraft were adequately predicted and were determined by the initial sounding also observed governing the Convective Available Potential Energy (CAPE), so again no model tuning was possible for the sake of a semblance of agreement. Over a dozen quantities were validated against the

coincident aircraft, ground-based and satellite observations (Fig. 6), including particle size distributions (Phillips *et al.* 2017b, their Figure 5). Raindrop-freezing fragmentation can be ruled out for that case as practically no supercooled rain was observed. Both models (AC, HUCM) fitted these observations 'like a glove', but only when including breakup in ice-ice collisions.

The aerosol-cloud model took as input the coincident observations (IMPROVE) of the mass concentrations of the 7 chemical species of aerosol. The CCN activity was predicted and then validated by Phillips *et al.* (2017b). The active IN was predicted from the observed dust, soot and organic concentrations by the empirical parameterisation (Phillips *et al*. 2008, 2013), which has been independently validated for various other coincident observations in our earlier papers. The CCN and IN activity spectra predicted by AC were then given to initialize the HUCM.

Finally, regarding an entirely different SIP mechanism, by pooling published lab observations of freezing drops in free-fall, a formulation was created for SIP by fragmentation of raindrops freezing quasi-spherically ('Mode 1'), (Phillips *et al.* 2018). By theoretical considerations, an alternative type of SIP from collisions of a raindrop with a more massive ice particle was represented ('Mode 2'). The formulation for both modes was applied in a detailed bin microphysics parcel simulation and shown to reproduce aircraft observations of cloud glaciation for a tropical convective case. Note that the SIP mechanism discussed by Phillips et al. (2018) has no connection whatsoever with the topic that it was erroneously cited for by KL2020, namely breakup in ice-ice collisions (see Corrigendum).

## 4. Comments on the Review Paper and its Corrigendum

### 4.1 Breakup in Ice-Ice Collisions

4.1.1 Criticisms by KL2020 of Both Lab/Field Experiments and of our Theories/Modeling

Including the revisions from their Corrigendum, the following statements were made by KL2020 (their Section 4) about fragmentation in ice-ice collisions, regarding both lab/field experiments by Vardiman (1978) and Takahashi *et al*. (1995), (Sec. 2) and our theoretical works.

Firstly, KL2020 state:

> *"Collisional ice fragmentation was also studied theoretically by ... Vardiman (1978) and Phillips et al. (2017a). These studies were based on the consideration of collisional kinetic energy and linear momentum. Such considerations would be relevant only for cases of direct central impact. In a general case, angular momentum and rotational energy should be taken into consideration. Since oblique particle collisions are more frequent than central collision, the efficiency of SIP obtained in these works is expected to be overestimated."*

Regarding the theoretical study by Vardiman (1978), it is evident that such negative criticisms are exaggerated, because our theory (Phillips *et al*. 2017a) is free-standing with a robust theoretical basis in principles of classical mechanics and statistical physics (Sec. 3). This allows us to quantify independently the errors inherent in published measurements from various simplifications in the design of both experiments (Vardiman 1978; Takahashi et al. 1995). Our theory relates numbers of
fragments to the fundamental determinative quantity, namely initial kinetic energy, for any collision. The errors, when quantified, transpire to be quite minimal in terms of numbers of ice particles emitted from any collision, for the following reasons.

In the above quote, KL2020 criticise Vardiman (1978) for failing to include rotational energy from oblique collisions in their
theory, their observations neglecting such collisions. Yet it can be shown that the final rotational energy is only a small fraction of the initial CKE. For any sphere of size, $R$, and mass, $M$, its moment of inertia is $I = (2/5)MR^2$. Consider a strongly oblique collision between two spheres of different sizes and assuming very frictional surfaces of both. During the rebounding part of the period of contact, the ratio of rotational to translational kinetic energy of the smaller sphere is approximated by that of a sphere rolling on a flat surface: 2/5. Here the assumed difference in size is sufficient to make the effective mass of the
pair approximately the same as the smaller particle; less than a 10% difference between effective and smaller masses implies a difference of a factor of more than 2 in size.   At the other extreme, for a head-on collision between such spheres there is zero rotation afterwards. This ratio of rotational to kinetic energies must be a monotonic function of the angle of impact, since any rotation only arises from torque exerted during impact. Thus, by this assumption of very frictional surfaces, the average fraction of initial CKE converted to rotational energy, accounting for all possible angles of collision, would be of the order of
10%. (For exact spheres, this must be an upper bound, since Supulver *et al.* [1995] observed for slow ice-ice collisions of 5-cm ice spheres in the lab that friction is insignificant, implying little transfer to rotational energy.)

The same is even more true of non-spherical particles. These may generally be approximated as ellipsoids. Similar arguments may be applied to an oblate ellipsoid (axial ratio of $0 < s \leq 1$) colliding with a much larger particle (by at least a factor of 2)
of any irregular shape. The smaller one falls with the longer axes horizontal initially (radius $a$). Now the non-spherical shape plays the role of perfect friction.   The moment of inertia of the smaller ellipsoid for rotation around a horizontal axis is $I = (1/5) M a^2 (1 + s^2)$. In the frame of reference of the centre of mass, the angular velocity of the smaller particle after the most oblique impact is $\omega \approx V/a$ where $V$ is its initial relative speed.  Thus, the ratio of rotational kinetic energy ($(1/2) I \omega^2$) after collision to the initial translational kinetic energy ($(1/2) M V^2$) is just $(1/5) (1 + s^2) \leq 2/5$, which reduces to 2/5 in
the special case of spheres ($s = 1$), the value noted above.   As before, the ratio for a head-on collision is zero.   Again the average fraction of initial CKE converted to rotational energy for all possible angles of collision is of the order of 10% and is smaller for oblate ellipsoids than for the spherical case. Alternatively, repeating this argument for a prolate ellipsoid ($s > 1$), $\omega \approx V/sa$ for the most oblique impact implies that a ratio of rotational (after) and translational (initially) kinetic energies is

$(1/5) \, (1 \, + \, 1/s^2) < 2/5$. The same estimate of the average fraction of initial CKE converted to rotational energy is reached (order 10%).

Thus, artificially preventing rotational evasive rebound in the lab/field experiment must have only weakly affected the energy available for fragmentation during impact (Eqs (1), (2)). This available energy is related to the difference between initial and final total kinetic energies. For example, a 10% change in the initial CKE would correspond at most to only about a 1% change in fragments emitted per collision ($N \approx 200$), in view of our analysis of observations by Takahashi *et al.* at various impact speeds (Phillips *et al.* 2017a), (Fig. 5). Thus, the omission of rotational evasive rebound from oblique impacts in Vardiman's theory, and in the observations of both lab/field experiments (Vardiman 1978; Takahashi *et al.* 1995), cannot seriously have biased the fragmentation per collision. In short, the error in either the measured or predicted numbers of fragments per collision, introduced by artificially preventing any rotational evasive response on rebound, is minimal, being about 1% or less. Naturally, these are theoretical estimates based on classical mechanics and such a lack of role for rotational evasive rebound for fragmentation awaits experimental confirmation.

Insofar as the above quote pertains to our theoretical formulation (Phillips *et al.* 2017a), it is wrong to suggest that our formulation somehow did not consider rotational effects or that it is somehow ill-posed by not treating rotation explicitly. Rotational KE after impact is included in Eq (1) underpinning our formulation and is only a small fraction of the total KE in any impact anyway, as proven below. As noted above (Sec. 3), the concepts of CKE (not momentum) and our energy-based definition of the coefficient of restitution are not restricted to head-on collisions. It is not true that our studies somehow required an assumption of either "*direct central impact*" or the concept of "*linear momentum*" for collisions in 1D. Our theoretical formulation treats collisions in 3D. Equally, the application of our theory to a simulation of clouds, after informing it with both lab/field experiments (Vardiman 1978; Takahashi *et al.* 1995) that involved head-on collisions with little rotational evasive rebound, merely introduces the same error to the predicted fragmentation that affected the observed numbers of fragments. As estimated above, that error was minimal (about 1% or less).

Secondly, KL2020 state:

> "The theoretical framework of collisional fragmentation developed in Yano and Phillips (2011), Yano et al. (2016), and Phillips et al. (2017[a]) was calibrated against experimental results of Vardiman (1978) and Takahashi et al. (1995) [Sentence A]. A detailed analysis of the Takahashi et al. (1995) laboratory setup indicated that the riming of ice spheres occurred in still air, which resulted in more lumpy and fragile rime compared to that formed in free-falling graupel. The collisional kinetic energy and the surface area of collision of the 2 cm diameter ice spheres also significantly exceed the kinetic energy and collision area of graupel whose typical size is a few millimeters. Altogether, it may result in overestimation of the rate of SIP, compared to graupel formed in natural clouds."

Sentence A would have been more accurate if it had mentioned that Yano and Phillips (2011, 2016) and Yano *et al.* (2016)

provided a theory of glaciation generally on the cloud-scale and did not focus on fragmentation at the particle-scale.   We did not use data from Vardiman (1978) for both theoretical studies.

There is the suggestion from the entirety of this quote from KL2020 (from Sentence A until *"overestimation …in natural clouds"*) that somehow simple curve-fitting of laboratory/field results was used to represent breakup in ice-ice collisions in

our theoretical and modeling work. That would be true if our theories somehow involved no dependency of fragmentation per collision on contact area or CKE such that all simulated graupel at millimeter-sizes in clouds would somehow have to fragment as if colliding just like the observed giant spheres (2 cm).  In fact, the opposite is true and our formulation of fragmentation in any given ice-ice collision (Phillips *et al.* 2017a) has a robust theoretical basis (Sec. 3).   The overall theoretical formulation itself is developed in a general sense from classical mechanics and statistical physics, as a versatile framework independent of

any particular laboratory experiment.  The fundamental quantities determining fragment numbers per collision are not merely particle size and temperature *per se*, but rather are the initial CKE, surface area for contact and the morphological classification of the ice (Eq (1)).  Such fundamental dependencies allow our formulation to apply universally to any size of colliding particle. It is not true that any "*overestimation of the rate of SIP*" must arise from the fact that *"2 cm diameter ice spheres also significantly exceed the kinetic energy and collision area of [most natural] graupel"*.  These fundamental dependencies (CKE,

surface area) of fragmentation are qualitatively realistic because they were derived by Phillips *et al.* (2017a) from universally accepted tenets of material science, such as the statistical distribution of asperities, and of classical mechanics, such as the law of conservation of energy in a collision and the concept of CKE (Sec. 3).  That law is part of the First Law of Thermodynamics.

Evidence for the realism of the mathematical form of our formulation of fragmentation is that the numbers of fragments per

collision observed by Takahashi *et al.* (1995) for a wide range of impact speeds (or CKEs) were realistically reproduced by our theory (Phillips *et al.* 2017a), (Fig. 5).  Naturally, Fig. 5 is not a validation since it merely shows comparison of our formulation with data used to constrain values of its parameters.  Yet equally, Fig. 5 is not the result of simple curve-fitting; rather it is the fitting of a simple theoretical expression to diverse observations at many impact speeds.  It fits the data 'like a glove'.   Moreover, additional evidence is from comparison with independent observations in Figures 4 and 6, as noted above

(Sec. 3).

In short, the fact that we fitted the theory to the giant spheres of the lab experiment of Takahashi *et al.* (1995) does not make it erroneous for smaller sizes, because our formulation (Eqs (1), (2)) has a sound theoretical basis on the conservation of energy and is universally applicable to particles of all sizes and morphologies.  CKE and surface area are always fundamental

quantities for breakup irrespective of the nature of the collision and the coefficient of restitution (defined in terms of energy) is an intrinsic property of the materials comprising the particles, as noted above.

Although the unrimed ice sphere in each collision observed by Takahashi *et al.* (1995) would be expected to have a "*fragile*" surface, they selected this combination of surface morphologies (rimed and unrimed, depositional and riming growth respectively) because their prior *in situ* observations of real clouds implied that it was representative for real in-cloud situations of ice multiplication. Previously, fall-out of large graupel into regions of weak LWC with small graupel had been observed *in situ* in winter-time clouds near Japan by Takahashi (1993) to generate copious ice crystals—with a novel video probe. High concentrations of ice crystals co-located with graupel had been observed in the mixed-phase region of deep convective clouds near Micronesia with the video probe flown on a balloon by Takahashi and Kuhara (1993), providing indisputable evidence of ice morphology and size. Such measurements by balloon are without the artificial shattering biases that afflicted airborne optical probes prior to 2011. The issue was then mitigated by improved probe design. Along its trajectory, LWC had earlier been sufficient to produce the graupel by riming and then was depleted (e.g. by riming), hence the weakness of the LWC there.

Consequently, Takahashi *et al.* (1995) argued that, in natural clouds, small graupel after forming through dominant riming growth can encounter predominant vapour growth when falling into such weak-LWC regions with LWC less than 0.1 g m$^{-3}$. Even if such parts of any convective cloud are limited, the in-cloud motions must tend to mix the crystals throughout most subzero levels of the cloud over time. The inherent nonlinearity of ice multiplication occurring by such breakup, involving growth of splinters to become ice precipitation by a positive feedback (Yano and Phillips 2011), must make such active parts of the cloud disproportionately influential compared to their volume.

Equally, the term "*calibrated*" in Sentence A seems misleading. A calibration usually suggests that a given model does not properly function without this procedure. That is hardly true here for any of our papers cited in that sentence (Yano and Phillips 2011; Yano *et al*. 2016; Phillips *et al*. 2017a). Our aim in these papers was not chiefly to provide a quantitative prediction of the exact time of a certain ice enhancement to be attained or of the exact number of fragments from a given collision of two particles of known size, morphology and ambient conditions. We were not providing merely a sort of metaphorical "speedometer" to infer the speed of ice multiplication. Rather our chief aim was to provide a theory based on principles of classical mechanics, material science and statistical physics with a versatile framework into which future lab observations may be assimilated as they arise. Our theories stand alone (Sec. 3).

Lastly, KL2020 claim that

> "*...No parameterizations of SIP due to ice–ice collisional fragmentation can be developed at that stage based on two laboratory observations, whose results are conflicting with each other.* [Sentence B]"

This claim (sentence B) has not been retracted about the impossibility of applying both lab/field studies for any formulation. When juxtaposed with sentence A, it effectively casts doubt on the integrity of our formulation (Phillips et al. 2017a) and theoretical studies (e.g. Yano and Phillips 2011) for no reason.     The fact that Phillips *et al*. (2017ab) constructed such a parameterization of breakup in ice-ice collisions, applied it in a model of real clouds by constraining its empirical parameters with both lab/field studies (Vardiman, Takahashi) criticized by KL2020 and then validated the simulation with aircraft data

(Fig. 6), proves that the statement is wrong.

    Moreover, both lab/field studies by Vardiman (1978) and Takahashi *et al*. (1995) differed in the microphysical species of ice they observed, with the only overlap between both being graupel/hail (Sections 2, 4.1.2). Most of the microphysical species observed by Vardiman (1978) were never studied by Takahashi *et al*. (1995). The deposition-*vs*-riming growth combination

of surface morphologies of the colliding pair (Sec. 2) in the indoor lab experiment by Takahashi *et al.* was never the objective of Vardiman, who observed particles outdoors. It is an exaggeration to claim both lab/field studies "*conflict*" with each other: they tackle different facets of a problem that is as complex as the morphology of ice itself.

    As justified above, both lab/field observational studies of breakup in ice-ice collisions (Vardiman 1978; Takahashi *et al.* 1995)

are sufficient to constrain representations of it in models of ice multiplication in natural clouds. Our formulation of breakup at the particle-scale (Phillips *et al*. 2017a) is based on fundamental principles such as conservation of energy, the fact that the coefficient of restitution is an intrinsic property of the materials of colliding particles and the statistical physics of asperities on their surfaces. This theoretical basis enables the formulation to be constrained by lab data even when the observational conditions of the experiment are not so representative of natural clouds (e.g. 2 cm sizes for Takahashi's experiment).


    The formulation was implemented in our cloud model, AC, with a storm simulation validated in many respects (Sec. 3; Fig. 6). The strong dependence of the simulated ice concentrations on the process of breakup in ice-ice collisions (Phillips *et al.* 2017b) makes this effectively a validation of the formulation itself. Equally, plausible errors in the assumed number of fragments per collision in the theory of Yano and Phillips (2011, 2016) and Yano *et al.* (2016) do not alter the fact that the

cloud-microphysical system remains unstable with respect to ice multiplication (Sec. 3; Fig. 3a).

    Finally, the theoretical results, from the 0D analytical model of cloud glaciation by Yano and Phillips (2011) and Yano *et al.* (2016), do not depend on laboratory results in any sensitive manner (Sec. 3; Fig. 3). A lab observation of $N$ is merely used for an estimate a value of the non-dimensional multiplication-efficiency parameter, $\tilde{c}$, along with several other values so as to

illustrate how natural deep convection is in an unstable regime with respect to ice multiplication. That investigation (Yano and Phillips 2011; Yano *et al.* 2016) was not fixed to this estimated value, but instead was performed over the full range of possible values of this non-dimensional parameter ($\tilde{c}$). In short, KL2020 give a misleading picture of our work, despite the Corrigendum.

### 4.1.2 Other Possible Criticisms of Both Lab/Field Experiments

Since sentence B (above quote) is such a strong and sweeping claim, it is valid to ask what other criticisms could be made to
support it beyond those expressed by KL2020. We identify here a few possible criticisms and then argue that they too lack credibility.

Firstly, one might criticize the fixing of the target (plexiglass or ice) in both experiments by Vardiman (1978) and Takahashi
*et al.* (1995). But this fixing can be shown to be unimportant for any energy-based formulation of fragmentation such as ours.
Essentially, since the target was rigidly fixed in both experiments, the effect on the collision is *via* a drastic increase of the
inertial mass of one of the colliding particles ($m_1$ and $m_2$). The "particle" becomes the planet Earth or effectively infinite
mass. The general equation for the initial CKE for a relative speed of $V$ (along the line of collision) is

$$K_0 = \frac{1}{2}\frac{m_1 m_2}{m_1 + m_2}V^2 \qquad (3)$$


Fixing one of the particles is expressed by $m_1 \to \infty$ and then $K_0 \to (1/2)m_2 V^2$. As an example, consider two identical
particles $m_1 = m_2 = M$ that collide when free. The initial CKE is then $K_0 = (1/4)MV^2$. Now we fix one of the particles,
and $K_0 = (1/2)MV^2$. Consequently, the effect from fixing one of the particles on the nature of collision is represented by
the CKE being doubled. Whether or not both particles are free does not alter the fact that CKE always provides the initial
energy relative to the centre of mass of the colliding pair, which is the source of energy for deformation (e.g. fragmentation).
CKE is always the fundamental quantity. Thus, since the formulation by Phillips *et al.* (2017b) related the number of
fragments to the CKE in the lab experiment, this relation is universal to any type of collision irrespective of whether or not
both colliding particles are free.

Indeed, the coefficient of restitution for head-on collisions of a pair of particles is observed, for a wide range of conditions, to
vary only weakly with their masses and impact velocity. For the present purposes, it may be viewed as an intrinsic property
of the materials of the colliding particles. For millimeter-sized ice particles impacting an ice wall, below a threshold of a few
metres per second of impact speed, the coefficient of restitution was observed to be constant, being independent of size and
impact speed (Eidevåg *et al.* 2021). That threshold (attributed by Eidevåg et al. to collisional melting) is larger than the impact
speeds observed by Vardiman (1978) and most of the speeds (about 1-4 m/s; Fig. 5) of the experiment by Takahashi et al.
(1995). Equally, Eidevåg *et al.* (2021) observed no significant effect on the coefficient of restitution for a 20-fold increase in

mass of a millimeter-sized ice sphere impacting an ice wall for any given impact speed below the threshold noted above. Whether the inertial mass of one of the colliding pair is infinite, by it being fixed, seems almost irrelevant to the value of the coefficient. This is why a fixed ice target sufficed in lab studies to observe the coefficient of restitution for free ice particles in planetary disks by Bridges *et al.* (1984), Hatzes *et al.* (1988) and Supulver *et al.* (1995). Moreover, when Takahashi et al. (1995) applied their lab data (sphere of 2 cm in diameter colliding with fixed target) to a collision between two spheres that are free of 0.5 and 4 mm in diameter, the ratio of the assumed masses is about 1000 and the CKE is practically the same as if one of them had been artificially fixed anyway. Consequently, the fixing of the target in both lab experiments introduces no significant error for our formulation (due to its fundamental basis on CKE) or for Takahashi's estimate for natural collisions between large and small graupel (applied by Yano and Phillips 2011).

Secondly, it could be argued that the plexiglass material of the fixed-plate used by Vardiman (1978), (Sec. 2), could have somehow biased the coefficient of restitution, altering the energy loss on impact and hence the fragmentation, relative to that expected for ice-ice collisions. However, Eidevåg et al. (2021) observed the coefficient of restitution to be practically the same when comparing ice-ice collisions and ice-ABS polymer collisions, both being in the range of 0.8 to 0.9 for impact speeds below the threshold noted above. Crucially, Vardiman (1978), before conducting the field campaign on the mountainside for the reported observations, compared the portable instrument with an alternative contrasting method (Sec. 2). Fragments were reported to be as numerous for impacts on the plexiglass (Fig. 1a) as for the copper mesh target (Fig. 1b).

Thirdly, it could be argued that somehow the weight of the fittings supporting the moving ice sphere in the Takahashi *et al.* (1995) experiment must have biased the measured fragment numbers. Indeed, inspection of published diagrams of their apparatus reveals a third ice sphere on the opposite end of the metal nearly horizontal rotating bar (about 13 cm long, 3 mm wide) for the main moving rimed sphere (Takahashi *et al.* 1995), (Fig. 2). It has the same size as the other two spheres but does not collide when they do. We estimate that altogether the moment of inertia for the main rotating sphere ("A" in Fig. 2) about the central axis supporting the bar, was approximately doubled by including this extra sphere and the weight of the moving fixtures. Since the total initial CKE equals the rotational kinetic energy of the ice sphere plus attachments (proportional to this moment of inertia) turning on the axis (almost vertical in Fig. 2), the initial CKE is roughly doubled by the extra mass. Hence, the extra weight for the rotating sphere acts to bias the measured numbers of fragments by only about 10% in light of observations by Takahashi *et al.* (1995) at various impact speeds (Fig. 5).

Fourthly, there is the criticism originally made by Phillips *et al.* (2017a) of the Vardiman (1978) experiment: ice particles were weakened by sublimation prior to collection outdoors, as they fell through the ice-subsaturated environment below cloud-base into the instrument on the mountainside where they broke up (Fig. 1), (Sec. 2). Phillips *et al.* reported that the prediction of $N$ from Eq (2), which for graupel-graupel collisions is fitted to only the Takahashi *et al.* (1995) observations, would be much lower (by about half an order of magnitude) than if Eq (2) were fitted only to observations by Vardiman (1978) for collisions

of graupel, for the same size (the largest he observed). Yet in reality this was never a grave problem for our formulation because Phillips *et al.* (2017a) then applied a correction factor to the fragility coefficient, $C$, in Eq (2) to yield a match at this size between both predictions. Phillips *et al.* then applied this same correction factor to the formulation for other microphysical species (graupel-snow and graupel-crystal collisions) when constraining them with the Vardiman (1978) data alone. Prior sublimation artificially boosting the observed fragmentation of natural ice falling onto the mountainside was invoked as the reason for this empirical correction by Phillips *et al.* (2017a).

Lastly, it could be argued that somehow the variation of rime density was not explored adequately in both lab/field experiments. Cloud-liquid properties and the impact speed of cloud-droplets when accreted were variables not varied by Takahashi *et al.* (1995). However, the standard conditions for riming in the experiment by Takahashi *et al.* (see Corrigendum of KL2020) are representative of those in which large graupel are typically created in clouds. When the duration of vapour growth and riming of both ice spheres was varied from 5 to 15 mins, the number of fragments measured by Takahashi varied by a factor of about 2.

In summary, contrary to sentence B of the review, in view of such errors, there is no evidence that the measurements of fragment numbers in ice-ice collisions by Vardiman (1978) and Takahashi *et al.* (1995) are either defective or unrepresentative such as to render them useless in estimating coefficients in a theoretical formulation. Naturally, these coefficient values can easily be improved as increasingly accurate laboratory measurements become available in future, but without changing the formulation itself in any manner.

### 4.2 Sublimational Breakup

On the topic of sublimational breakup, KL2020 conclude that

> "*this mechanism is also unlikely to explain explosive concentrations of small ice crystals frequently observed in convective and stratiform frontal clouds…Activation of SIP due to the fragmentation of sublimating ice requires* **spatial proximity** *of undersaturated and supersaturated cloud regions. In this case, secondary ice particles formed in the undersaturated cloud regions can be* **rapidly** *transported into the supersaturated regions..*".

The reason given by KL2020 is that ice fragments may disappear by evaporation before they can be recirculated. Although that is indeed a major limitation, in reality during descent there must be continual emission of fragments and their continual depletion by total sublimation. We argue that this causes a dynamical quasi-equilibrium with an enhanced fragment concentration. Hence, any mixing of downdraft air into the updraft will transfer air with the enhanced ice concentration into the updraft, whatever the depth or duration of the prior descent.

One can perform a scaling analysis as follows. Oraltay and Hallett (1989) observed rates of emission of the order of $F \sim 0.1$ fragments per second per parent dendritic crystal (a few mm) initially, during sublimation at relative humidities with respect to ice of about 70% or less. Such humidities would be attained in an adiabatic parcel descending at about 2 m/s from about -15 °C initially with about $n = 3$ L$^{-1}$ of crystals initially (2 mm). If each fragment takes a minute ($\tau$) to disappear by total sublimation, then the equilibrium number of fragments per parent particle is $F\tau \sim 0.1 \times 60 = 6$. Dong *et al.* (1994) observed rates of emission of $F \sim 0.3$ fragments per second per parent graupel particle (a few mm) initially. In a similarly subsaturated downdraft, there would be an equilibrium number of fragments per parent of $F\tau \sim 20$.

The pivotal point here is that such a quasi-equilibrium concentration is maintained throughout the entirety of the subsequent descent after being reached (e.g. for 10 mins of descent at 2 m/s). Thus, any recirculation of downdraft air into the surrounding convective ascent would transfer air enriched in fragments for their subsequent vapour growth and survival—irrespective of the timing of this recirculation during the descent and regardless of whether the prior descent is shallower or deeper.

To conclude, there is no reason to suppose that sublimational breakup is somehow insignificant in clouds, contrary to the impression given by KL2020.

## 5. **Conclusions**

The review of SIP by KL2020 (their Section 4) depicts our work (Yano and Phillips 2011; Yano *et al.* 2016; Phillips *et al.* 2017a) in a distorted manner, even after their Corrigendum. In particular, KL2020 made a controversial claim (sentence B above; Sec. 4) about the impossibility of developing any model formulation based only on the two existing experimental datasets of breakup in collisions of ice (Vardiman 1978; Takahashi *et al.* 1995). No quantitative evidence was provided by KL2020 in support of the claim. Our work proves that claim to be false: Phillips *et al.* (2017ab) constructed such a formulation of breakup in ice-ice collisions, applied it in a model of real clouds by constraining its empirical parameters with both lab/field studies (Vardiman, Takahashi) and then simulated a thunderstorm case with validation against aircraft data, with the formulation being critical for overall accuracy (Fig. 6). The formulation performed well.

That claim arose because KL2020 did not appreciate that our formulation of such breakup is based chiefly on an energy-based theory coming from theoretical physics that is universally applicable to all collisions (Phillips *et al.* 2017a). The truth is that this formulation can be delivered even without quoting the results from any laboratory experiments. KL2020 then supposed that any unrepresentative aspect of the collisions in the experiment will somehow cause problems when constraining parameters of any theory, when in reality our formulation is based on such fundamental quantities that this is not a problem.

In reality, the two laboratory/field studies of fragmentation may be applied to inform theories of fragmentation in any ice-ice collision, such as ours. As noted above, our formulation does not necessarily require such lab data anyway, as it stands alone on a sound theoretical basis. As far as we are aware, the only potential major issue with both lab/field experiments is a possible

bias from sublimation of natural ice particles outdoors prior to their fragmentation observed by Vardiman (1978). But Phillips *et al.* (2017a) knew about the possibility of such a bias and corrected for it when using both lab/field experiments to constrain some empirical parameters of their theory (Sec. 4.1.2). Anyway, the bias was not enough to alter the order of magnitude of $N$. It is a moot point whether the supposed sublimational weakening of these observations might actually have been representative of natural graupel aloft within clouds, since episodes of subsaturation with respect to ice (e.g. in ice-only

downdrafts) are likely along the trajectory of any graupel particle while in-cloud.

More crucially, the errors in the breakup rate per collision from the formulation, quantified by Phillips *et al.* (2017a) as a factor of about 2 or 3, soon become immaterial in the context of explosive multiplication of ice concentrations in a natural cloud (Yano and Phillips 2011, 2016). Whether the number of fragments is under- or over-estimated by an order of magnitude does

not alter the fact that explosive growth of ice concentrations by orders of magnitude will occur anyway *via* the positive feedback of ice multiplication (Sec. 3; Fig. 3).

For any ice multiplication somehow involving mixed-phase conditions (e.g. growth of fragments by riming to become graupel), the explosion occurs until an upper limit of the ice concentration is reached (related to onset of water subsaturation, depending

on temperature and ascent). This same final state is reached with little sensitivity to the breakup rate, with the timing depending only on the order of magnitude of the multiplication efficiency (Fig. 3). Both published lab/field experiments we used are sufficient to allow a formulation that produces a simulation of observed cloud properties including ice concentration with proven accuracy (Phillips *et al.* 2017b), as noted above.

Finally, the possibility of sublimational breakup contributing to observed ice enhancement cannot be dismissed as easily as the review paper suggests. In reality, a dynamical quasi-equilibrium is established between continual emission and total sublimation of fragments, so that any region of sustained subsaturation with respect to ice will develop an enhanced ice concentration persisting throughout the descent. The enhanced ice can then be transferred into regions of ascent subsequently for growth after almost any depth of descent—even after prolonged descent. An order of magnitude of ice enhancement is

possible from sublimational breakup in convective downdrafts and in their vicinity within the cloud. The quasi-equilibrium is sustained because the parent particles of ice precipitation take much longer to sublimate away totally than their fragments and continually emit fragments throughout most of their period of sublimation. The persistence of this quasi-equilibrium ice concentration was overlooked by KL2020.

In summary, there is no evidence that the only two lab studies about breakup in ice-ice collisions in-cloud hitherto (Vardiman 1978; Takahashi *et al.* 1995) are either mutually conflicting or erroneous such that they cannot be used to treat this process both in theoretical studies and numerical modeling of clouds, contrary to the claim by KL2020. There is no reason to suppose that those observations may cause any over-estimation of SIP in models.


**Acknowledgements.** VTJP was supported by three research grants, from the Swedish Research Council for Sustainable Development (''FORMAS'' Award 2018-01795), the Swedish Research Council (''VR''; Award 2015-05104), and the U.S. Department of Energy Atmospheric Sciences Research Program (Award DE-SC0018932).

**Data availability.** No datasets were used in the study.

**Author contributions:** VTJP conceived the study and wrote the comment with advice from JY. AD and DW advised about sublimational breakup and its role in clouds.

**Competing interests:** The authors declare that they have no conflict of interest.

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
