# Peer review of "Comment on "Review of Experimental Studies of Secondary Ice Production" by Korolev and Leisner (2020)"

_Atmospheric Chemistry and Physics, 2021_

## Referee Comment (RC1)

I'm openly sharing my identity. I have tried to address the authors' comment/concern of the Korolev and Leisner article which suggests that ice fragmentation during sublimation is not generally an important process for secondary ice production. To address that point, I drew on the Lagrangian spiral descents during the CRYSTAL-FACE field program, where the in-situ aircraft spiraled in a descending pattern, drifting with the wind and descending at a rate of about 2 m/sw. These were thunderstorm anvils, which began at a temperature of about -15C and ended at temperatures from about 6 to 8C. The relative humidity in the region at temperatures above 0C were significantly below 100% and the ice particles were sublimating, not melting. I have 3 figures for the 3 spirals, showing temperature, relative humidity and total ice concentration. I see no evidence of a concentration increase that would suggest ice fragmentation in the sublimating region is significant. Furthermore, in such regions, sublimating ice particle fragments would each rapidly fully sublimate or be collected by other particles. Andy Heymsfield

**CRYSTAL-FACE Lagrangian Spiral**

[Figure]

a: Temperature, Relative Humidity

b: Total Concentration > 30 microns

---

## Community Comment (CC1)

In support of Vaughan's hypothesis, I wanted to chime in with some evidence we have recently come across of potential sublimational SIP. Specific differential phase ($K_{DP}$) has recently been proposed as a potential marker of SIP zones as it is sensitive to large concentrations of anisotropic fragments, but this has so far been limited to SIP during riming (e.g., Grazioli et al. 2015; Sinclair et al. 2016; Kumjian and Lombardo 2017). However, looking at quasi-vertical profiles (QVPs) we have recently found bands of enhanced $K_{DP}$ situated squarely within the sublimation zone in 9 of 12 cases of strong sublimation, coincident with gradients of Z and RH w.r.t. ice (taken from coincident RAP analyses). We are currently exploring these signatures more quantitatively. However, given the observational laboratory evidence for sublimational SIP (e.g., Oraltay and Hallet 1989; Dong et al. 1994; Bacon et al. 1998), I contextualized these signatures with the existing literature by emulating Fig. 13 of Oraltay and Hallett (1989; reproduced below) and plotting the median $K_{DP}$ as a function of RHi and ice bulb temperature. Note that Oraltay and Hallett (1989) only included down to 50% RHi and -6C for ice bulb temperatures, so my figure is an extrapolation and it is unclear whether their 70% RHi threshold is maintained at much colder ice bulb temperatures. Regardless, it is apparent that these regions of enhanced $K_{DP}$ indicative of high concentrations of highly anisotropic particles are concentrated primarily in the region identified by Oraltay and Hallett (1989) as the favorable zone for sublimational breakup. The plot using mean $K_{DP}$ was similar. These sorts of values are not likely to be due to aggregates which we believe characterize the majority of parent particles in the cases examined, and riming is precluded due to the degree of dry air present. Thus, although the use of $K_{DP}$ is indirect evidence, under proper conditions (e.g., a continual and sufficiently high flux of parent particles into a sufficiently dry layer and favorable particle habits), I believe Vaughan's hypothesis of an equilibrium being established between fragment generation and fragment consumption to be plausible.

Jacob Carlin

[Figure]

Fig. 13. Display of the conditions for ice-breakup and water-shedding during evaporation and melting of dendritic ice crystals. Numbers of secondary particles per 5 mm crystal are indicated in parentheses.

(Reproduced from Oraltay and Hallett 1989)

---

## Author Response (AR1)

**Replies to Reviewer for "Comment on "Review of Experimental Studies of Secondary Ice Production" by Korolev and Leisner (2020)"**

1. **Comments from Referees**

Reviewer: This reviewer does not find the ideas and criticisms of KL2020 in manuscript particularly well organized or helpful with regard to evaluating the importance of ice-ice collisions to SIP.

Response:  We appreciate the effort made by the present reviewer to criticize the paper.  We have improved the organization of the manuscript.

However, the purpose of the commentary is not to evaluate the importance of ice-ice collisions for SIP in clouds.  That has already been established by Phillips et al. (2017ab) with detailed simulations of a mesoscale cloud system validated against aircraft data and other observations, applying a formulation of this breakup process. The ice enhancement by ice-ice collisional breakup was predicted to be almost 100.

**Phillips et al. (2017ab) provides details about the role of breakup in clouds.  Eventually the vast inaccuracy from omitting breakup in ice-ice collisions in current models will be widely recognised.**

Reviewer: The discussion is largely qualitative and a repeat of what has been published previously by the authors.  The language used is in several places exaggerated discussing the validation of both the Takahashi1995 experiments and the author's modeling results (presented in Yano [and Phillips] 2011, Yano2016, Yano2016b, Phillips2017, Philips2017b and Phillips 2018, hereafter termed "Y&P").

Response: The paper was Yano and Phillips (2011), not Yano (2011), and similarly for the papers in 2016. In all papers about this breakup, Phillips was a co-author and began the work on breakup a decade ago.

The second section of the paper by necessity must summarise the literature before starting the detailed criticism of the review by KL2020.   Anyway we have moderated the language in places to make it more cautious (e.g. "mistake" replaced by "misunderstanding").

**As we see it there are three points we wish to convey and we have updated the text to modify this. First, theoretical and modelling studies of SIP by breakup in ice-ice collisions are possible even without any further laboratory experiments.  Second, our previous theoretical and modeling studies of this type of breakup are valid EVEN IF all the previous laboratory experiments turn out to be totally erroneous.  Lastly, previous laboratory experiments about breakup in ice-ice collisions are, in fact, not as erroneous as KL2020 tries to suggest.   Any issues of representativeness or bias (e.g. sublimational weakening with Vardiman) in both lab/field studies are possible to correct for and are not prohibitively serious.**

[Figure]

**Figure A:** *(a) The phase-space of stability for the 0D model of ice-ice breakup for a population of ice crystals, and small and large ice precipitation particles. The multiplication efficiency, $\tilde{c}$, and dimensionless initial precipitation concentration are the two axes. In (b) the time evolution of the ice enhancement is shown for various values of $\tilde{c}$.  From Yano and Phillips (2011).*

Reviewer: The Y&P work makes an important contribution to SIP research and several papers are cited in KL2020 but not described in detail as the focus of the review is experimental studies of SIP. In the early papers Y&P use a single parameter, the number of fragments produced per collision, extracted (and scaled) from the Takahashi1995 laboratory experiments to characterize the fragmentation. The parameter in Yano [and Phillips] 2011 was not temperature, humidity, LWC or particle size/character dependent. **A more complex formulation is discussed in Phillips2017,2018 but this is not discussed here.**

Response: The commentary *does* discuss the Phillips et al. (2017a) paper in detail. The reviewed version of the manuscript has cited Phillips et al. (2017a) about twice as often as Yano and Phillips (2011) and even quoted the formula of the detailed formulation (Eq (2)) from that 2017 paper.

KL2020 (after the corrigendum) were claiming that our formulation could not be properly based on the Takahashi and Vardiman lab data. Our commentary is arguing that there is no real problem with the lab data.

Reviewer: At issue in this manuscript is the validity of the Takahashi1995 measurements and the simple scaling used in Yano [and Phillips] 2011,2016 to describe the in-cloud SIP process.

Response: The real issue at stake is the Phillips et al. (2017a) formulation because this is what KL2020 (after the correction) say should not have been based on the two sets of lab data.

Reviewer: Does the fragmentation observed by Takahashi1995 using 2 cm ice balls held on rods accurately simulate a SIP process in clouds? For several reasons I think this remains an open question.

Response: Yes, it clearly does reproduce it if two hail particles of that size collide in a natural cloud with the assumed history of growth (deposition and riming respectively).

In natural clouds, such hail particles (2 cm) grow by alternating episodes of dry and wet growth, hence the layered structure of hailstone sections. There is the re-freezing of the wet surface just like the wet surface of the frozen drop in the lab experiment. In the lab experiment of Takahashi, it was melted initially to free it from its metal mold, then with the surface re-frozen when placed in the cod-box, which seems qualitatively similar to the natural freezing of hail in-cloud.

Reviewer: I have the following comments on the manuscript:

Reviewer: 1) Line 11: Shouldn't the word be "untested" rather than "erroneous" in describing the current situation? It seems KL2020 is not suggesting Takahashi1995 is erroneous.

Response: We disagree. KL2020 are definitely suggesting that the Takahashi study is erroneous, as noted below.

KL2020 conclude about the Takahashi lab experiment: *"The collisional kinetic energy and the surface area of collision of the 2 cm diameter ice spheres also significantly exceed the kinetic energy and collision*

*area of graupel … Altogether, it may result in **overestimation** of the rate of SIP, compared to graupel formed in natural clouds".*

**There is the clear suggestion here from KL2020 that a grave error is introduced by applying the lab results to estimate the breakup of graupel in natural clouds, as Takahashi et al. (1995, their Section 4) were attempting to do. Such an estimate was the stated goal of their paper in 1995.**

Reviewer: We have no basis to conclude the validity one way or another. Instead, there is concern these results do not simulate actual in-cloud collision process. Confirming experiments with free-falling and proper-size ice particles have not been done yet.

Response: We disagree.

As simulated by Phillips et al. (2015), it is perfectly possible for 1-2 cm hail particles to form inside a cloud, and when they do so they will collide. The fact that in the lab one of the particles was fixed is equivalent to changing the CKE by a factor of 2 relative to both being free (as shown elsewhere here; Eq (3)) which is equivalent to a tiny error in the fragmentation rate (as shown in Section 4 of the commentary; Fig. 4). There is no problem provided one stratifies the data in terms of energy.

We agree plenty of experiments with more realism are a good idea. The only point we are making in the commentary is that the two published experiments from Vardiman and Takahashi allow the numbers of fragments per collision to be estimated, albeit imperfectly, as done by Yano and Phillips (2011), Yano et al. (2016) and Phillips et al. (2017b). Perfection is not needed for the effect from breakup to be simulated realistically, because there is nothing controversial about supposing that the initial kinetic energy in the frame of reference of the centre of mass of the two-particle collision is the source of energy for the fragments irrespective of whether both particles are free.

**Such energy conservation is an absolute constraint that all collisions, whether natural or artificial, must follow.**

Reviewer: There is a hint in the manuscript (see lines 145, 281, 288, 301) that the authors believe the comparison of their model results with field-study measurements (like in Phillips2017b) is in sufficient agreement as to validate a fit-derived fragmentation parameter and are not sensitive to the value extracted from Takahashi1995.

Response: It was not a mere hint; it was a definite statement.

We wrote: "*Both published lab/field experiments we used are sufficient to allow a formulation that produces a simulation of observed cloud properties in agreement with aircraft observations of ice concentrations and many other related properties (Phillips et al. 2017b)*" (line 301 of the reviewed version).

A comprehensive formulation was developed by Phillips et al. (2017a) with several dependencies on contact area, CKE, temperature, rimed fraction and the morphology of ice (snow, graupel, hail**). It was not a mere "fragmentation parameter" that was applied.**

Reviewer: If this is the claim, then this point is important and needs to be discussed in its own section.

Response: Yes, the point is important, but it is made amply by Phillips et al. (2017b). One only needs to read the abstract of the 2017b paper to appreciate it. It is not a mere claim that has been made. Rather, the formulation when applied in the model, is proven to yield agreement with the aircraft observations for many quantities.

**When a simulation is validated with innumerable cloud properties observed by aircraft, satellite, ground-based platforms and radar, after being initialized with coincident observations of 7 chemical species of aerosol and thermodynamic conditions, then this can be considered to be a proof of the accuracy of the model and its scheme for the leading sources of ice: breakup in ice-ice collisions. This is what our 2017b paper in Journal of the Atmospheric Sciences achieved.**

**The accuracy of the cloud simulation that included our formulation of breakup was proven by that 2017 paper (Fig. B). It was never a mere "claim". Only with breakup included is the observed ice concentration reproduced in the real storm (see red ellipse, Fig. Bd). We have now added extra text and a new Figure 5 to convey this point in Section 3 (lines 141-155), as required.**

[Figure]

*Figure B*:  *Validation of the aerosol-cloud (AC) simulation with and without breakup in ice-ice collisions, using aircraft data, radar data and satellite data.  Quantities shown are cloud-droplet concentrations and mean size, liquid water content, ice concentration, radar reflectivity, cloud fraction, ascent statistics and surface accumulated precipitation from (a) to (h).  Ice concentration is shown logarithmically in (d), both with and without breakup.  Another simulation of the same case by HUCM model is also shown.  Reproduced  from Phillips et al. (2017b).*

Reviewer: Phillips2017b shows good simulation-observation agreement but it is unclear to what extent the fragmentation parameter is the only free-parameter.  More discussion of how one can determine a fragmentation parameter from a SIP cloud model-field data comparison would be interesting.

Response:  **Fragmentation was not treated with a mere "parameter" by Phillips et al. (2017b).**

First, fragmentation was treated in detail with many dependencies according to theoretical formulation by Phillips et al. (2017) and based on the lab experiments by Vardiman and Takahashi.  Many empirically constrained parameters were involved.

Second, there were no tuneable parameters to adjust in our cloud model (Phillips et al. 2017b).  With most cloud models the CCN or IN activity is tuned to produce a semblance of accuracy.  But that was not true of our study.  The aerosol-cloud model (AC) takes as input the coincident observations (IMPROVE) of the mass concentrations of the 7 chemical species of aerosol.  The CCN activity was predicted and then validated by Phillips *et al*. (2017).  The active IN was predicted from the observed dust, soot and organic concentrations by the empirical parameterization (Phillips et al. 2008, 2013), which has been independently validated for various other observations in our earlier papers.  The CCN and IN activity spectra predicted by AC were then given to initialize HUCM.

Moreover, two cloud models with contrasting architecture simulated the same case (a mesoscale convective system 100 km wide) in the study by Phillips et al. (2017b). Ours (AC) has a hybrid bin/bulk microphysics scheme, the other was pure spectral microphysics (HUCM).  Both models allowed the same conclusion to be reached about the necessity for ice-ice collisional breakup in order for the observed ice concentrations and LWC to be reproduced.

Reviewer: 2) Line 22: "Impossible" seems an exaggeration.

Response:  No, it is not an exaggeration at all.

What we wrote was "*impression is given to the reader [by KL2020] that numerical modeling and theoretical studies of breakup in ice-ice collisions are somehow currently impossible due to the fact that reliable data for laboratory experiments are critically missing at present.*".

What KL2020 had actually written is "…*No parameterizations of SIP due to ice–ice collisional fragmentation can be developed at that stage based on two laboratory observations, whose results are conflicting with each other*".  Those two lab observations, namely by Vardiman (1978) and Takahashi et al. (1995), are the only ones to have been published so far and were named by KL2020.

**What we wrote was perfectly accurate as a summary of what KL2020 wrote.**

Reviewer: The question is whether the laboratory result and scaling used in Yano [and Phillips] 2011 is a realistic description of in-cloud processes.

Response: Well, the purpose of the Yano and Phillips 2011 paper was to provide a theory for hypothetically monodisperse populations of crystals, small and large precipitation with an idealized 0D

model.  Realism per se was not the primary goal.   As argued in the concluding section of the commentary, the value of the multiplication efficiency in the standard case (300) is so far above the critical threshold that whether the number of fragments is an order of magnitude too large or too small has little effect on whether explosive multiplication occurs.

Rather, it was the paper by Phillips et al. (2017ab) that provided a realistic description of in-cloud processes.  That realism was proven by separate validation of the scheme with independent data from Vidaurre and Hallett (2009), (Fig. 3 of commentary) and by the extensive validation for the real storm case observed by aircraft (Fig. 5 of commentary), (Phillips et al. 2017b).

Reviewer: 3) Line 23:  "unreliable, which is not the case," - this statement needs to be explained.

Response:  We now refer to the Section 4 in the commentary where evidence is summarized for the lab data being adequate for the purpose of constructing a formulation for an atmospheric model.

Reviewer: On what basis do the authors claim to know the Takahashi1995 data set is a reliable representation of in-cloud collisions?

Response: We know it is reliable because, in conjunction with the Vardiman observations, it was used to calibrate the general theoretical formulation of fragmentation for all types of collisions in natural clouds of Phillips et al. (2017a), which was then used in the validated simulation (**Fig. B**).  Also, the theoretical formulation by Phillips et al. (2017a) correctly reproduces the wide range of impact speeds observed by Takahashi et al. (1995).  Also the form of the formulation agrees with independent data from Vidaurre and Hallett (2009), (Fig. 3 of commentary).

Moreover, Vardiman observed fragmentation of graupel colliding with a fixed plate and after some correction, the formulation agrees with both that data and Takahashi et al (2009) for a given size, as explained by Phillips et al. (2009).

**Two lab studies (Vardiman and Takahashi) observed qualitatively the same phenomenon of breakup of graupel on impact.**

Reviewer: I have a similar comment to the text on Line 41 and several other places in the manuscript. Are the authors claiming a Phillips2017b-like analysis of other field data sets also shows good model-data agreement?

Response:  The storm simulated by Phillips et al. (2017b) is a mesoscale convective system half of the width of Texas.  The 3D domain was about 100 km wide.  Many convective cells and many cloud-types were present in the simulated storm.  The storm used for validation by Phillips et al. is representative.

The fact that so many quantities were validated by Phillips et al. (2017b), as noted above (**Fig. B**), means that this comprehensive validation suffices to demonstrate realism of the formulation of Phillips et al. (2017a).

**A Phillips2017b-like analysis of another field data set such as MC3E on 11 May shows adequate agreement and soon we will submit a paper about this.**

Reviewer: Wasn't the breakup parameterization used here more complex than simply the size, velocity, and KE scaling of Takahashi1995? Perhaps provide more discussion of the later model refinements and the evidence for the Takahashi1995 extracted parameter used in Yano [and Phillips] 2011?

Response:  Takahashi et al. (1995) rescaled the fragment number in terms of peak collision force, which they measured, rather than in terms of what we have always considered to be fundamental (CKE).   As we now say in the text (line 81), Takahashi et al. (1995) should have also rescaled it in terms of contact area.

Anyway, that issue does not affect the Yano and Phillips (2011) study much, since the multiplication efficiency is still in the regime of instability after reducing the fragmentation number by an order of magnitude for graupel-graupel collisions.

Reviewer: 4) Lines 52-110:  This material repeats much of what is stated in various places in Y&P.   It is so scattered and un-quantitative as to provide few new insights.  A more organized presentation would be appreciated.

Response: Those lines were not intended to provide new insights.  It is just a quick summary of the prior literature so as to set the scene for what follows.  Otherwise the reader may not follow what we are saying in the detailed counter-criticisms of KL2020's criticisms.

Reviewer: 5) I comment on the Takahashi1995 characterization:  This reviewer doesn't find the Takahashi1995 result particularly compelling as a simulation of what occurs between free ice particles in clouds.  Not to say it is erroneous, just on it's face not compelling.

Response: Let us take the opposite point of view:  suppose that somehow no fragments of ice are emitted from colliding graupel particles.  This is the implicit assumption of all cloud models to date (except ours).  The lab data by Vardiman and Takahashi are entirely inconsistent with that supposition. The notion seems most counterintuitive if one imagines hail colliding with graupel at high speeds of many metres per second in a cloud and if one considers the fact that rime can be fragile on graupel (Rango's SEM imagery) with a low bulk density (e.g. as low as 10% of pure ice).

**One is introducing grave errors if one omits any treatment of breakup in ice-ice collisions in a cloud model simply by speculating that the requisite lab data do not yet exist, as most cloud models implicitly assume.   Only by unrealistically treating IN activity can a bad model compensate for that.**

Reviewer:  The reviewer's opinion isn't particularly important here but again a more organized quantitative analysis of the Takahashi method rather than the collection of scattered hand-waving arguments would be appreciated if this is to be one focus of the manuscript.

Response:  Our arguments are not hand-waving at all.  Our theoretical formulation is published and has not been refuted.  The formulation rests on CKE being fundamental as the source of energy for fragmentation because energy is conserved and on the coefficient of restitution (related to fractional loss of energy) in an intrinsic property of the particle composition.

**In the commentary we estimate the errors introduced by various simplifications of the lab experiments and show these are negligible.   Our estimates are quantitative and rigorous.**

Reviewer: The SIP mechanism is unknown at this point.  There have been several suggested ideas.  The authors have published a model based on ice-ice fragmentation during collisions and claim their model can describe the process. The Takahashi1995 study collided two 1.8 cm diameter ice spheres, counted the crystals on a collector plate covering a fraction of the chamber bottom, and then multiplied that number by 4 as an estimate for the number of crystals ejected from their colliding spheres.  Are these crude experimental finding for 2 cm sphere collisions an accurate representation of the ejection rate for ice crystals in clouds?

Response:  Yes. This is because our universally valid theory has validated dependencies on the fundamental variables, namely CKE, contact area and temperature.

The experiment by Takahashi et al. (1995) was not crude.  If hundreds of splinters were emitted per collision, then the sample size is fine for extrapolation to all directions from the quadrant where they were counted.

**The reliability of the lab experiment is why our theory based on fundamental physics fits the lab observations by Takahashi for fragmentation over a wide range of impact speeds so accurately (Figures 3 and 4).  The lab data conforms what is expected from first principles.**

Reviewer: The scaling relation used in Yano [and Phillips] 2011 (to apply Takahashi1995 result to realistic cloud particles) considers only differences in particle diameter and fall velocity.  Later in Y&P kinetic energy, growth time, vapor density, collision dynamics/type and stochastic considerations are all mixed into this scaling.  But the scaling has not been confirmed by experiments.

Response:  No, that it is true.  Phillips et al. (2017b) did show comparison with observed fragmentation for a wide range of impact speeds both from Takahashi et al. (1995) and from Vidaurre and Hallett.

Reviewer: Korolev2010 mentions several questions in applying Takahashi1995 to actual cloud particle collision processes.  I won't describe these considerations but will discuss several other considerations.

Response: The reviewer means to say KL2020 ?

Reviewer: There are several ideas for how fragments occur during collisions.  Apparently for some temperature, LWC, convection and humidity/riming conditions, cloud processes produce irregular or "fuzzy" ice spheres with fragile irregularities protruding from their surface.  The idea presumes the protuberances grow with time and their fragility may (or may not) also increase.  When a collision occurs involving at least one of these fuzzy particles some protuberances break off as fragments.  This potential secondary ice production mechanism requires the fragments somehow find themselves a region with sufficient humidity to survive and grow thereby increasing the ice particle number density.

Response:  This is not a mere "*potential*" mechanism.  Its activity is now proven in our published papers for a real storm.  Phillips et al. (2017ab) created a theoretically justified formulation of breakup in ice-ice collisions, calibrated with published lab data (Vardiman and Takahashi) and then showed that when included in a detailed simulation of an observed storm its inclusion caused good agreement with many observations by aircraft (Fig. 5 of commentary; **Fig. B**).

Reviewer: Andy H's comment describes cases where the fragments likely will not survive.  Phillips2017b describes a case where the fragments apparently do survive.

Response:  Just to be clear, Andy H was commenting on a completely different mechanism for SIP, namely sublimational breakup.  This supposed mechanism has no connection with breakup in ice-ice collisions.   Andy H was addressing a different section of our review about that other mechanism, for which survival is the key issue.

Yes, in our simulations, we find breakup in ice-ice collisions occurs mostly in convective cores where there is mixed-phase conditions, such that the high humidity (water saturation) allows survival of fragments.

Reviewer:  Others will have different or more sophisticated ideas for the microphysics.  But in this SIP mode [Phillips et al. 2017b] the fragments originate at the surface of one or more of the colliding particles.  The surface of the particles involve roughening or new crystallite nucleation such that the protuberances grow via riming or vapor deposition.  These processes are temperature, RH, LWC and particle-size dependent.  The surface roughness of the spheres in the Takahashi experiments were not characterized.  At the surface during lumping/roughening or protuberance formation there will be epitaxial effects from the underlying crystallinity that likely will depend on the initial formation and growth process of the underly crystal.  The 2 cm spheres in Takahashi1995 began as frozen liquid water inside a metal sphere. **This freezing process is quite different from the variety of ice particle formation**

**processes that occur in clouds undergoing SIP.** Potential differences in the ice surface properties alone might call into question the relevance of the experiment to actual cloud particle-particle collisions.

Response:  **The reviewer raises a good point here about the variety of possible ways in which graupel or hail can form.  However, as we see it, the combination of Takahashi's (1995) experiment with the lab/field observations by Vardiman (1978) actually spans quite a variety of microphysical species of ice particles of diverse morphology.   In our 2017 formulation of breakup in ice-ice collisions we use both lab studies.  We now clarify this in a new Section 2.**

It is not true that the freezing process in Takahashi's lab experiment cannot happen in a real cloud.  Just as ice spheres were formed in the lab by freezing of water in a metal sphere released by slight melting of the exterior followed by re-freezing (Takahashi et al. 1995), also in a real cloud large graupel or hail particles are formed by densification during riming followed by alternating episodes of dry and wet growth.  These alternating episodes are the reason for the typical layered structure of translucent and transparent ice in sections through hailstones.  This is exactly the same process as in the lab experiment.

**There is no real problem here, as far as a pioneering study of fragmentation in clouds is concerned.  Of course, in future there will be many experiments that will relate the morphology of the ice surface to the fragmentation for each type of growth mechanism.**

Reviewer:  Second, the collision itself is likely different from what occurs between cloud particles.  In the experiment the 2 cm spheres move via a rod frozen into the center of the sphere.  The rigidity of the rod is important to the amount of energy exchanged in the inelastic collision.  A springy rod and axel will cause the balls to react much different than a ridged rod.

Response:  Yes, the rigidity of the rod is important for the energy exchanged and this differs from the situation in clouds.  No, that does not a problem for the energy-based formulation of fragmentation by Phillips et al. (2017a), expressed in terms of collision kinetic energy (CKE).

Since one ice sphere in Takahashi's experiment was fixed with a rigid metal rod, its inertial mass for the purpose of calculating the CKE was effectively infinite (mass of the planet Earth).  For such a situation the CKE reduces to be equal to the kinetic energy of the other ice sphere that is free, which is how Phillips et al. (2017a) calculated it when creating the formulation.

**Consider two identical balls of mass $M$ colliding with relative speed, V, head-on. When both are free, the CKE is $(1/4)\,MV^2$ .  When one of the balls is fixed, the CKE is doubled to become $(1/2)\,MV^2$. Consequently, effects from fixing one of the particles are included in the CKE.**

Always the CKE is the total kinetic energy in the frame of reference of the centre of mass of the two body system, irrespective of whether or not one of the bodies is fixed.  CKE is universally fundamental for the dynamics.

Whether one of the masses is small or large (e.g. the planet Earth when one particle is fixed) does not influence the coefficient of restitution (related to fractional loss of energy on impact), because it is an intrinsic property of the materials of the colliding particles and "*most of the kinetic energy dissipation occurs as a result of fracturing at the tiny contact surface area of the ice particles*" (Bridges et al. 1984). Indeed, this was why lab experiments by Bridges et al. (1984), Hatzes et al. (1988) and Supulver et al.

(1995) measured the coefficient of restitution using a fixed rigid target in the quest to study the ice particles in space (the rings of Saturn) that are free to rebound.  It is a truism that the same coefficient of restitution applies to head-one impacts of an ice sphere on a fixed ice wall as between two such ice spheres free to rebound.

**Always initial CKE is the source of energy for fragmentation as it is the total kinetic energy in the frame of reference of the centre of mass of the colliding pair, irrespective of whether both particles are free or one is fixed.  Phillips et al. (2017b) preempted the review's criticism succinctly: "*for head-on collisions the fixing of the target boosted the initial CKE without appreciably altering the energy-based coefficient of restitution q governing fragmentation*".    All of the analysis by Phillips et al. (2017ab) is in terms of CKE as the fundamental variable.**

Reviewer: Also during the collision, the strain along the axis of the rod will be different than in other directions causing perhaps larger amounts of gouging into the ice surface than would occur with free particles.  An apparatus holding the ice on rods adds complications to evaluating the forces in each inelastic collision. One suspects the energy transfer and the potential for gouging into the ice surface are different from what occurs when free-particles collide.

Response:  As noted above, the effect from fixing the ice particle is represented by an increase in the CKE.  It is still the CKE that is the source of energy for fragments, each of which requires a certain energy to form.

Reviewer: There are also aerodynamics considerations and charging effects for both the particles and the fragments created in colliding smaller crystals.  Perhaps all these potential effects wash out and the simple kinetic energy considerations are good enough to describe what is occurring.

Response: Aerodynamic effects are implicitly included in the CKE in our formulation (Phillips et al. 2017a) through the empirical fall-speeds (determining the impact speed) being determined by the drag force.

Most of the ice splinters are from the larger graupel (a few mm) colliding with snow (Phillips et al. 2017b).

Reviewer:  But it does seem fair that some experimental work using um scale and larger particles is needed before one can be confident the simple scaling idea, like that applied in the Yano [and Phillips] 2011 analysis, is valid.

Response:  **The Yano and Phillips (2011) analysis was theoretical.  The relevant paper to compare with experimental work is the detailed formulation by Phillips *et al.* (2017a).**

Reviewer:  This reviewer suggests the manuscript needs considerable re-work and clarification.

We have now done this.

**2. Author's Response**

Our point-by-point responses are included above adjacent to relevant points from the reviewer (Sec. 1).

Overall, the most important points of our commentary seem to have been missed by the review:

- **theoretical and modelling studies of SIP by breakup in ice-ice collisions would be possible even without any more laboratory or field experiments about it in future.**
- even if the Takahashi experiment were hypothetically to be found to be in error by an order of magnitude (it is not so erroneous in reality), our main theoretical conclusion (Yano and Phillips 2011) for a likely explosive multiplication would be unchanged due to the critical values being clearly away from the experimental value by orders of magnitudes (Fig. A):
  - for a change in the ice multiplication efficiency (proportional to the number of fragments per collision) by an order of magnitude, the time taken for an ice enhancement ratio of $10^2$ varies by only about 30 mins.
  - The standard value of Yano and Phillips (2011) for the multiplication efficiency is almost three orders of magnitude higher than the threshold for instability (unity), so such a change does not alter the fact that typically there is explosive growth of ice number.
- in fact, the Takahashi et al. experiment is sufficiently accurate for the sizes and types of the ice particles they observe and can be used to calibrate a theory for any natural sizes in-cloud

Consequently, it is wrong to argue, as done by KL2020, that no more theoretical and modeling studies are possible without further laboratory experiments

**3. Author's Changes in Manuscript**

From the reviewer comments, it is apparent that more background information is needed to assist the reader, primarily about the lab/field studies and our 2017 paper about simulation of an observed storm. So we have added new text by creating a new Section 2 and new figures (3 and 6). We describe in detail the model validation with the new formulation of ice-ice collisional breakup at lines 141-156.

As required in this review, we have moderated some of the language ("mistake" is replaced by a more diplomatic phase, "misunderstanding").

Finally, following the objection by the authors of KL2020 in the review process, we have removed the "personal communication" citation. As promised, we replaced this by a new subsection (Sec. 4.1.2) listing possible criticisms that are conceivable, which were informed by the discussions with the reviewer here and the authors off-line.

---

## Referee Report (RR1)

**Comment on "Review of Experimental Studies of Secondary Ice Production" by Korolev and Leisner (2020), Phillips et al., acp-123**

In this reviewers' opinion, the responses to the reviewers' comments are inadequate. How do you move the field forward-not by writing that there's little need to do more laboratory experiments-especially on fragmentation, . The study makes considerable use of the Takahashi (1995), experiments, where 2 cm ice balls were collided to generate fragmentations. How could that be considered the final answer? I have my own experience, working with a key player in recent identification of processes that lead to secondary ice production, and based on that have a theory that has not been examined in-depth.

The responses by Phillips et al. are highlighted below in bold text, my thoughts on the responses are not highlighted.

**Phillips et al. (2017ab) provides details about the role of breakup in clouds. Eventually the vast inaccuracy from omitting breakup in ice-ice collisions in current models will be widely recognised.**

Reasonable response, although I don't think they are correct in their interpretation.

**As we see it there are three points we wish to convey and we have updated the text to modify this. First, theoretical and modelling studies of SIP by breakup in ice-ice collisions are possible even without any further laboratory experiments**

I disagree with this point. It's an overconfident view

**Second, our previous theoretical and modeling studies of this type of breakup are valid EVEN IF all the previous laboratory experiments turn out to be totally erroneous. Lastly, previous laboratory experiments about breakup in ice-ice collisions are, in fact, not as erroneous as KL2020 tries to suggest. Any issues of representativeness or bias (e.g. sublimational weakening with Vardiman) in both lab/field studies are possible to correct for and are not prohibitively serious.**

This is not a good response in my opinion.

**In natural clouds, such hail particles (2 cm) grow by alternating episodes of dry and wet growth, hence the layered structure of hailstone sections. There is the re-freezing of the wet surface just like the wet surface of the frozen drop in the lab experiment.**

You haven't addressed the question in my opinion.

**There is the clear suggestion here from KL2020 that a grave error is introduced by applying the lab**

**results to estimate the breakup of graupel in natural clouds, as Takahashi et al. (1995, their Section 4) were attempting to do. Such an estimate was the stated goal of their paper in 1995.**

I agree with the reviewer and not the author

**As simulated by Phillips et al. (2015), it is perfectly possible for 1-2 cm hail particles to form inside a cloud, and when they do so they will collide. The fact that in the lab one of the particles was fixed is equivalent to changing the CKE by a factor of 2 relative to both being free (as shown elsewhere here; Eq (3)) which is equivalent to a tiny error in the fragmentation rate (as shown in Section 4 of the commentary; Fig. 4). There is no problem provided one stratifies the data in terms of energy.**

The vast majority of SIP production is in clouds where rimed particles and sometimes graupel are involved in the SIP process. The Takahashi 1995 study is not valid for the vast majority of situations.

**Such energy conservation is an absolute constraint that all collisions, whether natural or artificial,
must follow.**

This is an inadequate statement that is not proven with laboratory data. This is why more laboratory data are needed.

---

## Author Response (AR2)

**Summary of Author Responses to Reviewers**

It should be emphasized that neither of the reviewers point out anything to invalidate the merits of our commentary article. Thus, we believe that it must be accepted for publication, while also taking into account some of the comments of the reviewers.

Reviewer 1 only comments on secondary issues, and whatever way we adjust the comment text, these changes will not affect our main messages. The Reviewer 1 just needs to be slightly more concrete about the comments so that we can take them into account.

It appears to us that the Reviewer 2 rather misunderstands the basics of the scientific logic applied here. By our own interpretation, the comments by Reviewer 2 are based on those misunderstandings, and we find no basic validity. However, if we have misunderstood the comments, we sincerely request clear elaborations by this Reviewer.

In summary, we believe that the present comment must be accepted as it is, since neither Reviewer can provide comments that are mostly constructive.

**Author Response**

The following comments in the review, unfortunately, take the form of a collection of expressions of personal judgments, but without providing any supporting evidence for these judgments. More than often, stated judgments are hardly elaborated.  Thus it is just impossible for us to improve anything in response.

It is unfair that the review makes no attempt to provide any reasons when stating brief opinions.  We highlight this where it occurs below.

Any academic journal is a forum for reasonable debate. The review process normally includes such debate.  For any debate to be reasonable, reasons should be provided for opinions given.

**Point-by-Point Comments**

Reviewer:  In this reviewers' opinion, the responses to the reviewers' comments are inadequate. How do you move the field forward-not by writing that there's little need to do more laboratory experiments-especially on fragmentation ? The study makes considerable use of the Takahashi  (1995), experiments, where 2 cm ice balls were collided to generate fragmentations. How could  that be considered the final answer? Reviewer:  I have my own experience, working with a key player in recent identification of processes that lead to secondary ice production, and based on that have a theory that has not been examined in-depth.  The responses by Phillips et al. are highlighted below in bold text, my thoughts on the responses are not highlighted.

Response:  As this leading paragraph suggests, the present Reviewer is inherently skeptical about any theoretical studies unless there is overwhelming support from laboratory experiments and field observations.  We do not wish to change the Reviewer's personal perspective on this matter. We just need to point out that stand-alone theoretical studies are possible for the SIP as demonstrated by Yano and Phillips (2011), Yano *et al. (*2016) and Phillips et al. (2017a). Those studies demonstrate a potential possibility of an explosive SIP, that is of course, to be verified by observations.

In contrast with what the Reviewer suggests above, we have never stated at any place in our commentary article that "*there is little need to do more laboratory experiments*". Our own position is totally the opposite:  any healthy progress of science is possible only when both theoretical and modelling studies on the one hand, and the laboratory experiments and field observations on the other hand, are working together as equal partners. As KL2020 suggests in their conclusion, if we begin to argue that one side depends on the other totally, we are going to lose any healthy progress.

Naturally, we do not say that no progress is possible on one side, at all, without any progress of the other, which would mean that one side of the progress is only possible with the help of the other side. In fact, it is possible to launch a field campaign to look for a new mechanism of SIP, when there is only a very vague theoretical speculation to justify this.  Likewise, it would also be possible to imagine a

situation of performing some theoretical studies where, for whatever reason, there were to be only suggestive support from the laboratory experiments, or even just a very vague observational suggestion. Such stand-alone theoretical studies would have a role in the wider scientific context.

Finally, regarding what we actually wrote about sublimational breakup, there was quite a dramatic debate in the online interactive exchange. We clearly won the argument: KL2020 had claimed a short duration of sublimation is needed for fragments to survive, while we proved analytically that a quasi-equilibrium ice concentration arises that persists during lengthy convective descent. Regarding what we wrote about breakup in ice-ice collisions, in the last round we included many validation plots for aircraft data for storm simulations (Phillips et al. 2017b), deploying our formulation of this breakup in a cloud model (Fig. 6 of commentary). Only with the formulation included were the aircraft observations reproduced.

**It is unsurprising that the review makes no mention of this: these are both debates that we won.**

**Phillips et al. (2017ab) provides details about the role of breakup in clouds. Eventually the vast inaccuracy from omitting breakup in ice-ice collisions in current models will be widely recognised.**

Reasonable response, although I don't think they are correct in their interpretation.

Response: The Reviewer unfortunately chooses to disagree with us here. However, without any elaborations, we cannot further comment on this.

**Reviewer: …. First, theoretical and modelling studies of SIP by breakup in ice-ice collisions are possible even without any further laboratory experiments**

I disagree with this point. It's an overconfident view

Response: This is just a fundamental point: even without any laboratory experiments, certain theoretical studies are always possible. This does not mean to deny an importance of laboratory studies. We are just saying that it is wrong to suggest that the theoretical and modelling studies inherently depend on laboratory experiments: theories and modelling on the one hand, the laboratory experiments and field studies, on the other hand, are definitely mutually dependent, and it is rather unhealthy to suggest one side totally depends on the other.

**That was not an expression of overconfidence, but rather was a simple statement about the basic nature of theoretical and modelling studies.**

**Reviewer: Second, our previous theoretical and modeling studies of this type of breakup are valid EVEN IF all the previous laboratory experiments turn out to be totally erroneous. Lastly, previous laboratory experiments about breakup in ice-ice collisions are, in fact, not as erroneous as KL2020 tries to suggest. Any issues of representativeness or bias (e.g. sublimational weakening with Vardiman) in both lab/field studies are possible to correct for and are not prohibitively serious.**

This is not a good response in my opinion.

Response:  Although the Reviewer thinks that our previous response was not good, we cannot comment back on this without any reasons given.

**Reviewer: In natural clouds, such hail particles (2 cm) grow by alternating episodes of dry and wet growth, hence the layered structure of hailstone sections. There is the re-freezing of the wet surface just like the wet surface of the frozen drop in the lab experiment.**

You haven't addressed the question in my opinion.

Response: Here, again, the Reviewer chooses to disagree. However, we cannot comment  back on this without any reason given.

**Reviewer: There is the clear suggestion here from KL2020 that a grave error is introduced by applying the lab results to estimate the breakup of graupel in natural clouds, as Takahashi et al. (1995, their Section 4) were attempting to do. Such an estimate was the stated goal of their paper in 1995.**

I agree with the reviewer and not the author.

Response:  Again, it is difficult to respond to such tangential isolated comments when no coherent argument is expressed in the review.

**Reviewer: As simulated by Phillips et al. (2015), it is perfectly possible for 1-2 cm hail particles to form inside a cloud, and when they do so they will collide. The fact that in the lab one of the  particles was fixed is equivalent to changing the CKE by a factor of 2 relative to both being free (as shown elsewhere here; Eq (3)) which is equivalent to a tiny error in the fragmentation rate (as shown in Section 4 of the commentary; Fig. 4). There is no problem provided one stratifies the data in terms of energy.**

The vast majority of SIP production is in clouds where rimed particles and sometimes graupel are involved in the SIP process.

Response:  We agree.  The reviewer proves the point we are making:  the most prolific type of breakup in ice-ice collisions involves graupel, because graupel is dense and has the greatest CKE.   Phillips et al. (2017b) showed that graupel-snow collisions create the most ice fragments out of all types of collisions.

**Hail (defined as > 5 mm) is exactly the same general type of particle as graupel (defined as < 5 mm), with the only difference being that it is larger and hence denser.  Both types of particle grow predominantly by riming, hence the general densification as they become larger.   Density of accreted rime increases with fall-speed as the particle grows from being graupel to hail.**

**In none of our theoretical studies do we actually apply to all sizes of graupel the exact fragmentation number measured for hail-sized ice spheres.**

Reviewer: The Takahashi 1995 study is not valid for the vast majority of situations.

Response:  We disagree.   Yes, most clouds displaying SIP do not have 1-2 cm hail.  No, that is not a problem because such clouds do usually involve graupel and graupel is the same general type of particle as hail, as noted above.  And our theoretical formulation (Phillips et al. 2017b) is universally applicable to all sizes of particle, so that fitting its parameters to the Takahashi results allows it to be then applied to collisions among graupel/hail of all sizes.

**Reviewer: Such energy conservation is an absolute constraint that all collisions, whether natural or artificial, must follow.**

This is an inadequate statement that is not proven with laboratory data. This is why more laboratory data are needed.

Response:  This is an extraordinary claim by the review.

The context for our quoted statement in the last round of responses was: "*Perfection is not needed for the effect from breakup to be simulated realistically, because there is nothing controversial about supposing that the initial kinetic energy in the frame of reference of the centre of mass of the two-particle collision is the source of energy for the fragments irrespective of whether both particles are free*".

We never wrote that energy conservation is the only law of conservation to apply to a collision nor that it is the only constraint, nor did we write that the actual magnitude of the initial kinetic energy is somehow independent of whether both particles are free (we wrote that it is not).  We were writing about the general principle of conservation of energy, which is part of the First Law of Thermodynamics.

**The law of conservation of energy is ineluctable as a basic principle of science.  It states that the total energy of an isolated system is constant.  This needs no experimental verification.  It is the basis for the entire field of classical mechanics in Physics for the last few hundred years.**

**Here, the present Reviewer asserts that the adoption of the energy conservation principle here is "inadequate", and insists that it must be proved by laboratory data.  Of course, such an insistence is just unreasonable: throughout the history of science, it is common knowledge that no physical principle was ever "proved" by laboratory data, which is always imperfect.  Only after enough "supports" (these hardly constitute "proofs") by laboratory data do scientists decide to accept these principles.**

**Replies to Reviewer 2**

**Author response**

We are grateful to the reviewer for their effort in scrutinizing the manuscript.

**Point-by-point Responses**

Reviewer:  The authors claim that 2 statements in the review article by Korolev and Leisner (hereafter KL2020) are misleading and distort the validity of their contributions.

*Statement 1: The theoretical framework of collisional fragmentation developed in Yano and Phillips (2011), Yano et al. (2016), and Phillips et al. (2017[a]) was calibrated against experimental results of Vardiman (1978) and Takahashi et al. (1995).*

Here what the authors said in their articles:

P.2 of Yano2011 states, "The goal of the present article is to demonstrate the important efficacy of this mechanical breakup (or fragmentation) process by a theoretical investigation. For this purpose, we take the parameters estimated by more recent laboratory data (Takahashi et al. 1995)."

P.2 of Yano2016 states, "Yano and Phillips (2011) and Yano et al. (2016, here- after YP11 for the former and YP collectively) show, under a deterministic approach, by taking the experimentally estimated parameters by Takahashi et al. (1995), that the ice breakup process can indeed lead to explosive ice multiplication under certain regimes."

P12 of Phillips2017a states, "Theoretically unknown parameters are estimated from the observations, both from outdoors and laboratory experiments, by Vardiman (1974, 1978) and Takahashi et al. (1995)."

Although the word "calibrated" is not used in the text, it does characterize pretty-well what the authors state in their publications.

In this manuscript, one of the main points is to lessen the stated connection between their modeling results and the Vardiman1978 and Takahashi1995 experimental results. The authors claim their modeling study results stand despite the possibility that the experimental results could be 'totally erroneous'.

I think this is correct and perhaps a better characterization of the connection is that the modeling results are consistent with these previous laboratory experiments. This idea is stated well in the manuscript. But this idea is more of a correction and corrigendum to the author's previous statements than it is a comment on Korolev2020.

Response**:**

We never wrote that the experimental results from Vardiman and Takahashi could really be totally erroneous, nor do we believe this to be even a real possibility.   We were merely mentioning a purely imaginary scenario, for the sake of argument ("*Secondly, hypothetically our previous theoretical and modeling studies would still be valid even if all the previous laboratory/field experiments about fragmentation in ice-ice collisions were to be shown to be totally erroneous*").

In criticizing our comment concerning the Statement 1, we are afraid to say that the review fails to convey any understanding about the difference between "calibrations" to adjust the model and simple "estimations" of values.   A calibration usually suggests that a given model does not properly function without this procedure.   That is hardly the case here for any of the papers cited (Yano and Phillips 2011; Yano et al. 2016; Phillips et al. 2017a).

Both Yano and Phillips (2011) and Yano *et al*. (2016) provide theoretical analyses that do not require calibrations from any laboratory experiments. The behavior of both versions of the model is defined solely in terms of a single nondimensional parameter, $\tilde{c}$. Both analyses were performed over a full possible range of $\tilde{c}$ without referring to any laboratory experiments. That is exactly what we mean by both theoretical studies are 'stand-alone'.  Although this point was not explicitly stated in the original articles, it must have been obvious for all the readers who already know how to interpret the theoretical studies.  We are afraid to say that both the authors of KL2020 as well as the present Reviewer do not understand the basic nature of theoretical studies.

As these three quotations above show, we do refer to those laboratory experiments for the purpose of estimating the value of this nondimensional parameter, $\tilde{c}$.  However, the theory itself does not need this specific number. The sole purpose of getting the number is to infer where a typical atmospheric cloud is situated along the coordinate of this nondimensional parameter so that a link between the theory and the real world can be established in a solid manner.   However, even if (that is a purely hypothetical situation) new laboratory experiments in future, somehow, were to provide completely different estimates, then there would be no need for us to repeat those theoretical studies again: the theory part would stand by itself, because our existing studies are not based on mere "calibrations" as KL2020 wrongly assert.

Similarly, Phillips et al. (2017a) created a theoretical formulation of the numbers of fragments per collision by deriving an expression from the law of conservation of energy.  An energistic coefficient of restitution was applied.  A statistical distribution of the strength of asperities was derived from other observations to create the expression.  The main thrust of the study was provision of a versatile framework into which future lab observations could be assimilated.   For an application to real simulations, some of parameters were inferred from observations by Vardiman (1978) and Takahashi et al. (1995).

Thus, the above sentence coupled with the very next three sentences in KL2020 together give a false impression that somehow the formulation by Phillips et al. (2017a) was obtained by simple curve-fitting to lab data and would have never functioned properly without both lab/field studies.   Moreover, when the formulation is applied in a cloud simulation, as argued in our commentary there is a lack of sensitivity of the eventual ice concentrations to errors in the formulation since the simulated cloud system is in the explosive unstable regime anyway.

Reviewer: *Statement 2: No parameterizations of SIP due to ice–ice collisional fragmentation can be developed at that stage based on two laboratory observations, whose results are conflicting with each other.*

It is not clear how this statement applies to the author's work since they claim their model formulation and result is not based or dependent on the experimental work. They claim their work is valid even if the experimental work is 'totally erroneous'. The authors explain this well and this warrants publication as it emphasizes a point not well stated in their previous publications. But it would be nice of the authors also were to add some new results.

After describing the Vardiman1978 and the Takahashi1995 results, KL2020 claim it is hard to judge the consistency of the results given the differences in experimental setups and conditions. I agree but they then go on to state these results are conflicting. I agree with the authors that the word "conflicting" is likely not a proper characterization of the 2 laboratory studies. One has c within the range 1 -100 and the other has c = 50 (which may be corrected due to contact area corrections as the authors suggest, to 5-10). The authors make several good points about their model results being consistent with the experimental results. But they also concede that the laboratory experiments could be totally erroneous. So, while it is fair to disagree with the KL2020 characterization, all this hand waving will not be settled without more laboratory work.

The KL2020 review was not focused on modeling efforts. Phillips2017b cloud model-field data comparison shows good agreement with a fragmentation parameter consistent with the laboratory experimental results. This work certainly stands on its own. I suggest the authors reconsider the title of the manuscript. Much of the manuscript stands on its own as a clarification of what was previously stated in the author's publications.

Response:  The statement 2 here is simply logically wrong: it is not true that neither theoretical nor modelling progress is possible without any further laboratory experiments.  Even in the absence of those two laboratory/field experiments (Vardiman, Takahashi et al.), our theoretical studies (Yano and Phillips 2011, Yano et al., 2016) would have been possible as already emphasized above. Our modelling development has also been possible, thanks to already available knowledge form the studies in statistical physics, although with uncertainties in specifying some model parameters.

**We are not sure how to respond to the actual comments by the Reviewer, who half agrees with us, but also half disagrees with us.  One wonders if there is any fundamental disagreement here.**

Reviewer:  Smaller points:

Line11-16: Please break this into 2 sentences. There is also a typo here. Also be careful with the word 'valid' - usually this means validated by experiment. But I'm having trouble following the logic: How can the theory or model be validated by experiment when, at the same time, the authors concede the experimental results could be totally erroneous? You can't have it both ways can you?

Response:  There was a misunderstanding of what we wrote here.  We have never validated our any application of our theoretical formulation with the lab data used to constrain any of its parameters (Vardiman, Takahashi *et al*.).

**No, 'valid' does not necessarily mean validated by comparison with observations.  Rather the technical term is "validation" or "to validate", as in "model validation".**

**Anyway the sentence has been re-phrased and condensed (line 15).**

Reviewer: Line 107-108: 30 minutes is not the case for all \tilde{c}. Time is much longer for the range \tilde{c} \sim 1 to 10 as shown in Fig 3. Perhaps mention what is the lowest value of \tilde{c} consistent with the time-scale observed for the clouds in the field observations?

Response, lines 127-131:  Done.

Reviewer: Line 179: The word 'consistent' rather than validated?

Response:  No, the term "*validated*" is fairly applied everywhere.  We will not remove the term.

"Model validation" refers to the technical procedure of comparing a model prediction with coincident independent observations, after using observations of the case to initialize the model.  Figure 6 shows many "validation plots".

**Phillips et al. (2017b) use the term "validation" for this rigorous comparison with aircraft observations.**

Reviewer: Line 265: Better to state the limit rather than 'is minimal'.

Response, line 272:  Done.

Reviewer: Line 268: The 1% is only true for a certain range of values given the curve

Response:  Yes, it is the value for the slowest impact speed observed by Takahashi et al.  The other impact speeds they observed have an even weaker percentage change than this (< 1%) because they observed the fragmentation to approach an upper limit as the speed increased.

**Text is now clarified (line 267).**

---

## Author Response (AR3)

Replies to Editor's Comments

**Summary of Author Response**

We are grateful for the positive verdict.

We agree that constructive debate with sharing of ideas is always a good idea for progress in science. One could even argue that certainty in science is never possible and that scientific progress involves simply continual shifts in consensus as new observations, models and ideas arise and as fresh generations of scientists enter the arena. Discussions are the essence of science.

We agree that future lab observations will no doubt change and clarify the ideas from the past two lab studies that we have been debating the merits of, perhaps in ways we cannot yet imagine.

We have tried to re-phrase our ideas better to clarify our position and to provide counter-arguments, as required.

**Point by Point Responses**

Editor: After reviewing your responses to the reviewers and conferring with referee #2, I am generally inclined to accept your commentary for publication. However, there are still a couple of points that you would like considering rephrasing. This may be in part because different communities, here experimentalists and modelers, have different connotations of the applied phrasing.

Response: We agree, there seem to be slightly different philosophical perspectives at play among the various communities. They all have different fortes and unique contributions to make.

Editor: In the revised version of the manuscript, you make clear that the underlying theory of breakup in ice-ice collisions stands alone. I think there is no doubt about this valid contribution to this research area. However, as it is written, I agree with referee #2, that how you use Vardiman 1978 and Takahashi et al. 1995, is best described as calibration.

Response: In the manuscript, our objection is not so much about the use of the term "calibration", but rather is the suggestion from the entirety of the quoted four sentences that somehow a simple curve-fitting was applied. The notion of the particles being unnaturally large in Takahashi's experiment (2 cm) is followed by the suggestion that this will cause any "calibrated" theory to over-estimate the fragmentation. That would be true if no dependence on impact speed were included in the theory and it was simply related to contact area. But that is not the case for our 2017 formulation, which was informed by both lab/field experiments (Takahashi, Vardiman).

We think the term "calibration" induces the reader to view our studies as somehow providing a "speedometer of ice multiplication". One calibrates an instrument that will not function without the calibration. Our studies are much more than that.

**To clarify our position, we added fresh text at lines 323 and 369-376.**

Editor: You derive that for c > 1, you have explosive breakups. With no experimental data at hand, which c value would you chose? For example, for a c =1, you may discard the entire effect (if I am not mistaken). c=300 is likely too much, etc. To give you an example what I mean: a speedometer by itself is fundamentally valid, i.e., a linear response to a rotating wheel. However, it is only as good as the calibration is. As the measurement of speed gets better, the accuracy of the speedometer also increases (assuming its theory of operation is correct). Again, this has nothing to do with the foundations of the speedometer/theory. My feeling is that no one doubts fundamentally the theory you developed and applied. However, there are many theoretical studies in our field that need further laboratory data to yield higher accuracy. This is a typical process.

Response:  **Yes, if $c < 1$ then the theory becomes superfluous, as we argued here.  But that is not likely since the errors in both lab studies of fragmentation are limited, as evinced by Section 4 of the commentary.  Takahashi's own videosonde observations in convective clouds informed the design of his lab experiment.**

Essentially, the process we are describing is fundamentally nonlinear.  There is an explosive super-exponential growth of the ice concentration towards the maximum possible for mixed-phase conditions with the positive feedback elucidated by Yano and Phillips (2011) and simulated in detail by Phillips et al. (2017):  ice crystals grow to become small ice precipitation (snow or graupel) and then large ice precipitation (graupel), with continual collisions generating ice fragments that grow as crystals, ad infinitum.   It is the order of magnitude of $c$ rather than $c$ itself that determines the time-scale of the explosion.   So, the analogy of the speedometer would be more applicable if the log of $c$ corresponded to the "speed" that determines the time of the journey.

Yet, as we argue above, a metaphorical "speedometer" is not the aim of our theoretical studies.  We are not providing a mere device for measuring the rate of ice multiplication.

We have two independent lab studies, measuring different types of microphysical species in collisions, and they both report appreciable fragmentation.  For our detailed simulation of a cold-based convective storm, for which we showed excellent and comprehensive validation (see Figure 6 in the commentary; Phillips et al. 2017a), we estimated that the explosive growth of ice concentration corresponded to $c = 10$ in our model, albeit from breakup in graupel-snow collisions rather than with the graupel-graupel collisions assumed by Yano and Phillips (2011).   Phillips et al. (2021, in review) show sensitivity tests with the same simulation and in all the perturbation simulations for varied cloud-base temperature, solute and solid aerosol conditions, updraft speed etc, we find generally the ice-ice breakup boosts the order of magnitude of the ice concentration as the dominant mechanism of SIP among those mechanisms represented in the model (H-M, raindrop-freezing fragmentation, breakup in ice-ice collisions).

**In summary, it is the order of magnitude of $c$ rather than $c$ itself that is important for the cloud glaciation, and both lab studies we base our estimates of fragmentation on are not so erroneous as to make c greater than unity when it should be less or *vice versa.***

**Note that KL2020 claimed that both lab studies are "conflicting" but they never provided any quantitative evidence to support that claim.**

Review:  I also feel that the wording in the review article of both studies being conflicting may not be the ideal choice of words. Likely they are not consistent with each other since they were operated a different conditions. However, in my opinion, this is not such an important point. Only more studies will help to achieve convergence and this process usually takes time.

Response:  Agreed.   The term "conflicting" was unsubstantiated by any quantitative evidence from KL2020.   Not only were the conditions of the collisions different (different sizes, different humidities) between both studies but also the microphysical species studied were, mostly, different:

- Vardiman:  lightly, moderately and heavily rimed snow crystals, both dendritic and non-dendritic; graupel
- Takahashi:  rimed graupel vs graupel in depositional growth

Editor:  Regarding the point of failing to include the rotational energy from oblique collisions. I appreciate your discussion on this matter. Your reasoning makes sense. However, and I believe both referees struggled with this, you describe a highly idealized case of spheres. In the atmosphere, we are likely not dealing with spheres but with irregular shaped ice crystals. I can follow your argument in the manuscript, but would it stay around 10% for rotating ellipsoids colliding or other shapes? Of course, you argue for the case of the experiment where spheres were present, but the results of your theory are applied to the "real" world where you likely have not perfect spheres.

Response:  Yes, we estimate that for other non-spherical shapes the ratio of final rotational to initial translational kinetic energies would be of the order of 10% or less.

The only material assumption is that the difference in sizes is such that the effective mass of the colliding pair is approximately the same as that of the smaller particle.  In other words, the difference in size is at least a factor of 2.

**We have added text (lines 273 – 284) to extend our argument for ellipsoids of any shape.  The "real world" consists of ice particles whose shapes, albeit irregular, may be approximated by prolate (columnar crystals) or oblate (planar crystals, snow, graupel, hail, freezing drops) spheroids.**

Editor: I do not think that this issue needs to be further discussed in detail here but maybe acknowledging caveats, when data is lacking, may be appropriate. In other words, your discussion may imply that this effect is "always" negligible although we have never "observed in situ" two colliding particles followed by breakup. Someone operating in the atmosphere (not lab or model) may misunderstand your points and will just disagree instead of seeing the broader application of your theory. Hence, providing caveats may be beneficial, also to motivate more research.

Response: The caveat in the context of the rotational rebound issue is added as required (lines 295-296).

Editor: Referee #2 and I feel this is valuable addition. I hope these thoughts help to perform the last minor revisions before publication of your manuscript.